# Neural-HSS: Hierarchical Semi-Separable Neural PDE Solver

## Abstract

Deep learning-based methods have shown remarkable effectiveness in solving PDEs, largely due to their ability to enable fast simulations once trained. However, despite the availability of high-performance computing infrastructure, many critical applications remain constrained by the substantial computational costs associated with generating large-scale, high-quality datasets and training models. In this work, inspired by studies on the structure of Green's functions for elliptic PDEs, we introduce Neural-HSS, a **parameter-efficient** architecture built upon the Hierarchical Semi-Separable (HSS) matrix structure that is provably **data-efficient** for a broad class of PDEs. We theoretically analyze the proposed architecture, proving that it satisfies exactness properties even in very low-data regimes. We also investigate its connections with other architectural primitives, such as the Fourier neural operator layer and convolutional layers. We experimentally validate the data efficiency of Neural-HSS on the three-dimensional Poisson equation over a grid of two million points, demonstrating its superior ability to learn from data generated by elliptic PDEs in the low-data regime while outperforming baseline methods. Finally, we demonstrate its capability to learn from data arising from a broad class of PDEs in diverse domains, including electromagnetism, fluid dynamics, and biology.

## 1 Introduction and Related Work

Machine learning is emerging as a powerful tool for accurately simulating complex physical phenomena (Brandstetter et al., 2022; Li et al., 2021; Boullé et al., 2023). Unlike traditional numerical methods, which rely on explicit mathematical models and require problem-specific implementations for spatial resolution, timescales, domain geometry, and boundary conditions, machine learning models learn directly from data—either from simulations or real-world observations—enabling greater flexibility and scalability. Moreover, these models leverage ongoing advances in GPU acceleration and large-scale parallelization, with continued improvements in both accuracy and efficiency.

A wide range of architectural primitives has been explored for modeling different physical systems, each leveraging structural biases that arise in the solution operator of specific classes of PDEs. As prominent examples, the Fourier neural operator (Li et al., 2021) learns a kernel integral operator through convolution in Fourier space, enabling efficient representation of global interactions; message passing neural networks (Brandstetter et al., 2022) capture localized interactions via graph-based message updates; U-Net ConvNets (Gupta & Brandstetter, 2022) exploit multiscale representations to couple fine- and coarse-scale features; while operator transformers (Hao et al., 2023) leverage transformers to handle challenging settings such as irregular meshes. Other methods directly approximate the action of linear operators in the elliptic setting (Boullé et al., 2023), bypassing explicit discretization of the underlying equations. Moreover, Boullé et al. (2023) show that the underlying learning problem is well-posed for a very small number of datapoints and prove that the class of elliptic PDEs is "data efficient": the number of training points needed to learn the solution operator depends only on the problem dimension. This effect is due to the nature of the solution operator for elliptic-type PDEs, which is highly structured: the Green function $G(x, y)$ associated with the solution operator is a Hierarchical Semi-Separable (HSS) mapping that exhibits low rank when restricted to off-diagonal subdomains.

In fact, the Multipole Graph Neural Operator employed in (Boullé et al., 2023) is motivated by a hierarchical structure observed in Green functions (Boullé & Townsend, 2024, Section 2.4), known as the Hierarchical Off-Diagonal Low-Rank (HODLR) structure. In an HODLR matrix, off-diagonal blocks at different scales are well-approximated by low-rank factorizations. When used within a neural architecture, this flexible representation allows the network to capture both near-field and far-field interactions in a multiscale fashion and is one of the key factors behind the data-efficiency property of the neural architecture.

The HSS structure is a special case of the more general HODLR format, a hierarchical domain-decomposition strategy that has been widely studied in the scientific computing literature for developing fast and memory-efficient solvers for elliptic-type PDEs, e.g., (Martinsson & Rokhlin, 2005; Gillman et al., 2012). Compared to HODLR, the HSS model enforces a nested basis across levels in addition to low-rank off-diagonal blocks. This nested parameterization greatly reduces redundancy, yielding a more compact representation and faster matrix–vector products, improvements that are not available in the plain HODLR structure.

In this work, we propose Neural-HSS, a neural architecture that injects this type of structure into a neural network model for PDE learning. Thanks to the efficient data representation of HSS, the proposed model is able to approximate solution operators with far fewer parameters than baseline models while retaining better or comparable accuracy.

The geometric structure of HSS operators imposes an inductive bias that concentrates modeling capacity on local interaction effects in the physical system—modeled through full-rank submatrices—while approximating interaction effects between distant subdomains using low-rank matrices, resembling a mean-field approximation. This modeling strategy underlies many popular neural PDE solvers. In fact, architectures such as ResNets (He et al., 2016), Swin Transformers (Liu et al., 2021), and Message Passing Neural Networks (Gilmer et al., 2017) devote the majority of their representational capacity to modeling local interactions. This bias aligns well with the nature of PDE dynamics, in which the dominant behavior is driven by localized interactions, in contrast to integro-differential equations, which can incorporate global interaction effects. The effect of this implicit bias was also highlighted in (Holzschuh et al., 2025), where the authors demonstrate that in a Swin-Transformer-based model, increasing the attention window size leads to a rapid performance plateau.

Overall, our main contributions are as follows:

- We propose Neural-HSS, a **parameter-efficient** neural architecture with a novel type of layer inspired by HSS theory for PDEs (Hackbusch, 2015), and we establish its universal approximation property.

- We prove that the proposed architecture is **exact** and **data-efficient** on the class of elliptic PDEs, i.e., (a) the global minimizers of the empirical loss represent the discretized solution operator exactly, and (b) the number of data points required to learn the exact operator depends only on the intrinsic dimensionality of the problem. To the best of our knowledge, this is the first result for an architecture that can handle arbitrary types of PDEs while guaranteeing exactness and data efficiency for a broad class of PDEs.

- We highlight an intriguing connection between the proposed architecture and a discretized version of the convolutional-type layer used in FNO architectures, showing that an HSS layer can approximate it arbitrarily well with very few parameters.

- Under the assumptions of Theorem 2.3, which match the setting of (Boullé et al., 2023), we conduct two experiments in 1D and 3D. The 3D experiment is performed on a grid with 2M points, a well-known challenge for machine learning models. In both cases, we demonstrate the superior performance and scalability of Neural-HSS.

- We conduct extensive experiments on a broad class of PDEs arising from electromagnetism, fluid dynamics, and biology, showcasing strong performance relative to commonly used architectural primitives such as ResNet and FNO layers in terms of parameter efficiency, test error, and computational time. In particular, this last advantage becomes more evident for higher-dimensional PDEs. Moreover, we demonstrate the effectiveness of Neural-HSS beyond elliptic PDEs, showing that the same architecture is also effective for different classes of nonlinear PDEs.

**Related Work.** There has been a surge of interest in learning-based approaches for improving classical solvers for linear PDEs. Several works focus on accelerating iterative methods such as

Conjugate Gradient for symmetric positive definite systems (Li et al., 2023a; Kaneda et al., 2023; Zhang et al., 2023), or GMRES-type solvers for specific applications like the Poisson equation (Luna et al., 2021). Others focus on learned preconditioning strategies: neural networks have been used to construct preconditioners that speed up convergence (Greenfeld et al., 2019; Luz et al., 2020; Taghibakhshi et al., 2021), or to optimize heuristics such as Jacobi and ILU variants (Flegar et al., 2021; Stanaityte, 2020). NeurKItt (Luo et al., 2024), for example, employs a neural operator to predict the invariant subspace of the system matrix and accelerate solution convergence. However, these approaches are primarily designed to enhance classical linear numerical pipelines. In contrast, our work takes a fundamentally different perspective: we aim to learn an end-to-end solver.

Moreover, recent work has focused on learning low-dimensional latent representations of PDE states for efficient simulation, extending classical projection-based reduction (Benner et al., 2015) with deep learning methods such as autoencoders (Wiewel et al., 2019; Maulik et al., 2021), graph embeddings (Han et al., 2021), and implicit neural representations (Du et al., 2024; Chen et al., 2022). Koopman-inspired methods enforce linear latent dynamics (Geneva & Zabaras, 2022; Yeung et al., 2019), while latent neural solvers and transformer-based ROMs have been developed for end-to-end modeling (Li et al., 2025b; Hemmasian & Barati Farimani, 2023). In addition, Kissel & Diepold (2023b) introduces a hierarchical Fan et al. (2019b) network to model the nonlinear Schrödinger equation. Another key challenge remains long-horizon stability, motivating strategies like autoregressive training, architectural constraints, and spectral or stochastic regularization (Geneva & Zabaras, 2020; McCabe et al., 2023; Stachenfeld et al., 2022). More recently, generative models, in particular using diffusion-based models, have shown promise for stable rollout and data assimilation by producing statistically consistent trajectories (Shysheya et al., 2024; Li et al., 2025a; Andry et al., 2025).

The use of structured matrices in neural network architectures is also highly relevant and used across different areas of deep learning. Recent approaches include the use of low-rank (Schotthöfer et al., 2022; Zangrando et al., 2024) and hierarchical decompositions (Fan et al., 2019a; Kissel & Diepold, 2023a), low-displacement rank (Thomas et al., 2019; Zhao et al., 2017; Choromanski et al., 2024), butterflies and monarchs (Fu et al., 2023; Dao et al., 2019; 2022).

## 2 SETTING AND MODEL

In this section, we present the necessary definitions and theoretical motivations for our proposed architecture. We start with the following formal definition of HSS structure:

**Definition 2.1.** (HSS structure (Casulli et al., 2024, Definition 3.1)) Let $\mathcal{T}$ be a cluster tree of depth $L$ for the indices $[1, \ldots, d]$. A matrix $A \in \mathbb{R}^{d \times d}$ belongs to HSS$(r, \mathcal{T})$ or simply HSS$(r)$ if there exist real matrices

$$\{U_\tau, V_\tau : \tau \in \mathcal{T}, \, 1 \leq \text{depth}(\mathcal{T})\} \quad \text{and} \quad \{D_\tau : \tau \in \mathcal{T}\}$$

called *telescopic decomposition* and for brevity denoted by $\{U_\tau, V_\tau, D_\tau\}_{\tau \in \mathcal{T}}$ or simply $\{U_\tau, V_\tau, D_\tau\}$, with the following properties:

1. $D_\tau$ is of size $|\tau| \times |\tau|$ if $\text{depth}(\tau) = L$ and $2r \times 2r$ otherwise;

2. $U_\tau, V_\tau$ are of size $|\tau| \times r$ if $\text{depth}(\tau) = L$ and $2r \times r$ otherwise;

3. if $L = 0$ (i.e., $\mathcal{T}$ consists only of the root $\gamma$) then $A = D_\gamma$;

4. if $L \geq 1$ then $A = \boldsymbol{D}^{(L)} + \boldsymbol{U}^{(L)} A^{(L-1)} (\boldsymbol{V}^{(L)})^T$ where

$$\boldsymbol{U}^{(L)} := \text{blkdiag}(U_\tau : \tau \in \mathcal{T}, \text{depth}(\tau) = L), \quad \boldsymbol{V}^{(L)} := \text{blkdiag}(V_\tau : \tau \in \mathcal{T}, \text{depth}(\tau) = L),$$

$$\boldsymbol{D}^{(L)} := \text{blkdiag}(D_\tau : \tau \in \mathcal{T}, \text{depth}(\tau) = L)$$

and the matrix $A^{(L-1)} := (\boldsymbol{U}^{(L)})^T (A - \boldsymbol{D}^{(L)}) \boldsymbol{V}^{(L)}$ has the telescopic decomposition $\{U_\tau, V_\tau, D_\tau\}_{\tau \in \mathcal{T}_{2r}^{(L-1)}}$, where $\mathcal{T}_{2r}^{(L-1)}$ denotes a balanced cluster tree of depth $L - 1$ for the indices $[1, \ldots, 2^L r]$, see Definition A.2.

This definition is a natural extension of the idea of a low-rank linear operator, when the matrix is not globally low-rank but admits a low-rank expansion on subdomains that do not intersect.

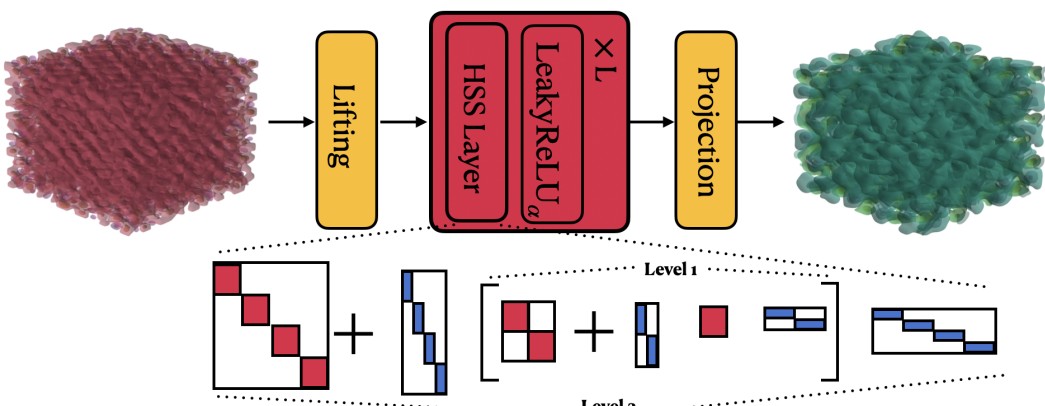

Figure 1: **Model overview.** The lifting and projection layers can be implemented either as HSS layers or as full-rank layers. We illustrate, as an example, the weight matrix structure with two different hierarchical levels.

The utility and efficiency of these structures become immediately clear when considering elliptic-type PDEs. In fact, when one discretizes an elliptic partial differential operator—say by finite differences, finite elements, or boundary integral methods—a large algebraic system $Au = f$ arises, where $A$ is associated with discretized Green's functions. It turns out that many of the off-diagonal blocks of $A$ have low numerical rank because of the smoothing properties of elliptic operators, making distant interactions "weak" and therefore inducing a decay in the singular values, which depends on the regularity of the kernel (Bebendorf, 2000; Bebendorf & Hackbusch, 2003). HSS matrices exploit exactly this feature by organizing the matrix into a hierarchical block structure in which off-diagonal blocks are approximated by low-rank factorizations. Thus, HSS-based models are used to design fast numerical solvers or preconditioners for elliptic PDEs by efficiently approximating the operator, thereby reducing storage and computational complexity (Börm, 2010; Börm & Grasedyck, 2005; Gillman et al., 2012). As the HSS structure is invariant under matrix inversion, these properties demonstrate that HSS operators can effectively model the structure of the solution operator and offer a powerful modeling primitive for a neural PDE solver.

Based on this key observation, we propose below the Neural-HSS model.

## 2.1 MODEL OVERVIEW

In this section, we present the proposed Neural-HSS architecture, as illustrated in Figure 1.

**One-dimensional HSS layer.** For 1-dimensional problems, the HSS layer consists of an HSS structured matrix followed by a nonlinear activation. In Appendix D we describe the forward pass in more detail, along with input and output channels of the layers, the number of levels, and the rank of the layer. The rank controls the size of the low-rank coupling matrices between sub-blocks, enabling efficient compression of off-diagonal interactions. The number of levels determines the recursion depth, i.e., how many hierarchical splits the input undergoes. Clearly, the HSS layer is fully compatible with the backpropagation algorithm. The structure of the layer allows us to stack multiple layers, enabling the construction of deeper architectures.

**Activation function.** We employ the $\text{LeakyReLU}_\alpha$ activation. $\alpha$ is typically a tunable hyperparameter; however, in our implementation, $\alpha$ is learnable. In particular, if the underlying PDE is linear, the model adapts $\alpha \to 1$, effectively recovering the identity function. Instead, for nonlinear PDEs, $\alpha$ can deviate from 1 to capture the nonlinearity where necessary. This activation function is also relevant because it fulfills the exactness guarantees presented in Theorem 2.3. While this is the activation of choice in our implementation, we emphasize that this choice is not restrictive: other activation functions can be used if better suited for the problem at hand. We also highlight that as long as the activation acts entrywise, the HSS-induced structural bias is maintained as the topology of the interlayer connections is not affected.

$m$**-dimensional HSS layer.** The complex hierarchical architecture of HSS operators in higher dimensions poses significant challenges, as it depends on the geometry of a partition of the domain, which can be arbitrarily complex. For this reason, different possible models can be used to extend the HSS structure to higher-dimensional tensors (Hackbusch, 2015, Chapter 8). Here we extend the HSS layer to higher-dimensional PDEs by parametrizing each layer as a high-dimensional tensor obtained as a low-rank expansion of the outer product of one-dimensional HSS layers. More precisely, an $m$-dimensional HSS layer is a tensor of outer CP-rank $r_{\text{out}}$ parametrized as follows:

$$H_\theta^m : \mathbb{R}^{\overbrace{d \times \cdots \times d}^{m}} \to \mathbb{R}^{d \times \cdots \times d}, \quad H_\theta^m(\mathcal{Z}) := \sum_{k=1}^{r_{\text{out}}} \mathcal{Z} \bigtimes_{j=1}^{m} W_j^{(k)} \tag{1}$$

$$W_j^{(k)} \in \text{HSS}(r_{k,j}) \subset \mathbb{R}^{d \times d}, \quad \theta = (W_j^{(k)})_{j,k} \text{ trainable parameters}$$

where the definition of modal product $\bigtimes$ is recalled in Definition A.1 and $\mathcal{Z}$ is the layer's input tensor. This model is motivated by the effectiveness of the Canonic Polyadic decomposition in representing very high-dimensional tensors with a small number of variables (Hitchcock, 1927; Lebedev et al., 2015). In fact, while the parameter count on a generic linear map between tensors with $m$ modes would scale as $O(d^{2m})$, the memory for this $m$-dimensional HSS layer scales as $O(r_{out}mrd)$ when $r_{k,j}$ are all equal to $r$. Thus, the architecture's parameter-efficiency increases as the dimensionality of the PDE grows.

## 2.2 THEORETICAL RESULTS

In this section, we present our main theoretical results, showing that the proposed Neural-HSS enjoys some useful properties.

First of all, we notice in the next Theorem that the HSS structure is efficient in representing convolutional-type operators in which the kernel is regular. Thus, each HSS layer in the proposed model can be interpreted as a generalization of a (discretized) convolutional-type linear layer, such as those implemented in popular FNO or CNO architectures (Li et al., 2021; Raonic et al., 2023), as well as classical convolutional filters.

**Theorem 2.2.** *(Convolutional kernels are HSS approximable) Let $D \geq 1$, $\Omega \subseteq \mathbb{R}^D$ be a compact set, let $k : \Omega \to \mathbb{R}$ be an asymptotically smooth convolutional kernel (Definition A.4). Consider the operator*

$$T : \mathcal{C}^0(\Omega; \mathbb{R}) \to \mathcal{C}^0(\Omega; \mathbb{R}), \quad (Tf)(x) = \int_\Omega k(x - y) f(y) \, dy$$

*For a set of basis functions $\{\phi_j\}_{j=1}^K$, consider the discretization matrix*

$$A_{ij} = \int_\Omega k(x - y) \, \phi_i(x) \, \phi_j(y) \, dx \, dy, \quad i, j = 1, \ldots, K.$$

*Then, for every $\eta$-admissible pair (Definition A.3) of well-separated clusters $\tau, \tau'$, there exist $r = O(\log(1/\varepsilon)^D)$ such that*

$$\inf_{B \,:\, \text{rank}(B) = r} \|A|_{\tau \times \tau'} - B\|_F \leq \varepsilon$$

*where $A|_{\tau \times \tau'} := \Pi_{\tau'} A \Pi_\tau$ is the orthogonally projected operator in the subdomains spanned by the nodes $\tau, \tau'$.*

The result in Theorem 2.2 shows that each regular convolutional operator on well-separated domains can be well approximated by a low-rank matrix, where the rank grows logarithmically with the tolerance. In other terms, if the domains are well-separated and the kernel is regular enough, then interactions between subdomains can be well-approximated through a low-rank expansion. Thus, the whole convolution can be approximated in $\text{HSS}(r)$ by recursively partitioning the index set into clusters. The proof of Theorem 2.2 is included in Appendix B.

Next, we analyze approximation and data efficiency properties of the model. Note that by setting the number of hierarchical levels to its minimum and simultaneously increasing the rank to its maximum, the backbone effectively reduces to a standard multilayer perceptron (MLP). This observation immediately implies that Neural-HSS inherits the well-known **universal approximation** property of MLPs, at the cost of possibly increasing the number of levels of the tree and the rank.

Moreover, in combination with piecewise linear activation functions with learnable slope, it satisfies data-efficiency and exactness recovery properties, as formalized in the following:

**Theorem 2.3.** *(Exact Recovery and Data-Efficiency) Consider the model*

$$\mathcal{N}_\theta(b) = H_\ell \circ \cdots \circ H_1(b), \quad H_i(z) = \text{LeakyReLU}_{\alpha_i}(W_i[z]), \quad \theta = (W_1, \alpha_1, \ldots, W_\ell, \alpha_\ell)$$

*and for $\lambda > 0$, the loss function together with its set of global minimizers*

$$\mathcal{L}_N(\theta) = \sum_{i=1}^{N} \|\mathcal{N}_\theta(b_i) - u_i\|_2^2 + \frac{\lambda}{2} \sum_{i=1}^{\ell} (\alpha_i - 1)^2, \quad \mathcal{M}_N := \underset{W_i \in \text{HSS}(r_i, \mathcal{T}), \alpha_i \in \mathbb{R}}{\arg\min} \mathcal{L}_N,$$

*where the datapoints $\{(b_i, u_i)\}_{i=1}^{N} \subseteq \mathbb{R}^{d^k} \times \mathbb{R}^{d^k}$ are standard-Gaussian distributed and satisfying $\mathcal{G}u_i = b_i$ with $\mathcal{G}^{-1} \in \text{HSS}(r, \mathcal{T})$. Furthermore, suppose $\max_i r_i \geq r$.*

*Then there exists a constant $C > 0$ such that, whenever $N \geq C \cdot \sum_{i=1}^{\ell} r_i$, the following exact recovery identity holds with high probability*

$$\sup_{\theta \in \mathcal{M}_N} \left\| \mathcal{N}_\theta(\cdot) - \mathcal{G}^{-1}(\cdot) \right\|_{L^\infty} = 0.$$

*Consequently, any global minimizer $\theta^* \in \mathcal{M}_N$ has zero generalization gap, i.e., $\mathcal{N}_{\theta^*} = \mathcal{G}^{-1}$ and the model is data-efficient as exact recovery is possible with a number of data points $N$ that depends only on the architecture's intrinsic dimensionalities $r_i$.*

A proof of Theorem 2.3 is included Appendix C. This result shows that even in possession of only a possibly very small number of examples, if the underlying operator is HSS, then all global minimizers of $\mathcal{L}_N$ recover the exact solution operator. This is, for example, the case for linear elliptic PDEs. Moreover, notice that Theorem 2.3 formalizes the "data-efficiency" for the proposed architectures, in the spirit of what was done in (Boullé et al., 2023). An experimental validation of this result is presented in Section 3.1 and Figure 3, where we numerically compare the data-efficiency of our architecture against that of other baseline models.

## 3 EXPERIMENTS

In this section, we will present the numerical results. For a more complete description of the PDE problems considered and the data generation, we refer to Appendix F; for the hyperparameter settings and model implementation details, we refer to Appendix N; for more details on the training loss and evaluation metrics used, we refer to Appendix G.

### 3.1 DATA EFFICIENCY

We conduct two experiments analogous to those in (Boullé et al., 2023). The first experiment focuses on the one-dimensional Poisson equation, where we train our models on datasets of varying sizes, ranging from 10 to $10^3$ samples. The second experiment considers the three-dimensional equation on a grid with resolution $128 \times 128 \times 128$, corresponding to approximately $2 \times 10^6$ grid points. Here, we vary the training set size from 16 to 256 samples. We use fewer training samples in three dimensions because, as shown in (Boullé et al., 2023), meaningful conclusions can already be drawn with training sets of around 200 samples. Moreover, generating three-dimensional data and training models on it is computationally expensive, further motivating the need for data-efficient models in 3D simulations.

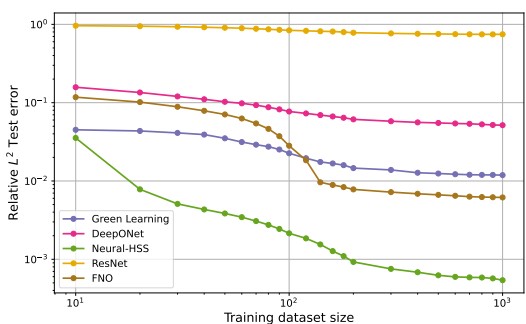

Figure 2: Train size vs relative test error for different models. The models are trained on a 1D Poisson equation.

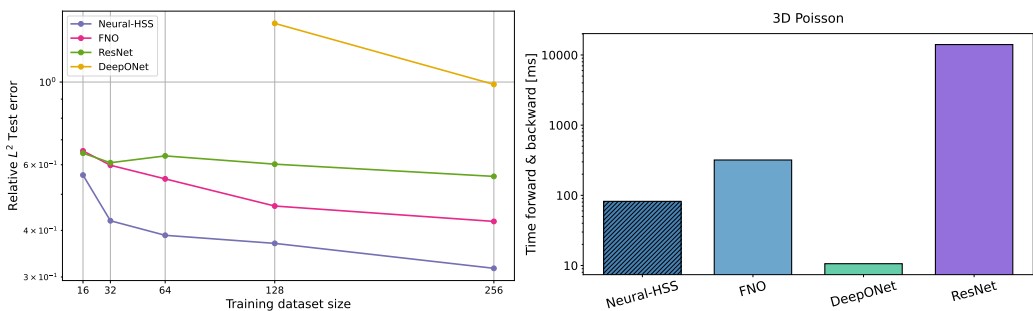

Figure 3: **Left**: Train size vs relative test error for different models. The models are trained on a 3D Poisson equation. **Right**: Timing of forward and backward for the different models.

In the 3D experiment in Figure 3, we do not
report the performance of the Green Learning model (Boullé et al., 2023), since the memory of an NVIDIA A100 80GB was not sufficient already using a batch size of 1 and a depth of 1. We also omit DeepONet results for training set sizes smaller than 128, as its performance in this regime was poor and including it would clutter the visualization. The full plot can be found in Appendix H. Details on the models are provided in Appendix N.

**1D** Our findings in Section 3.1 are consistent with those reported in (Boullé et al., 2023). With a limited training sample budget, Green Learning outperforms the FNO model, as observed in (Boullé et al., 2023); however, as the number of samples increases, FNO scales more effectively and surpasses Green Learning once the training size exceeds $10^2$. In contrast, Neural-HSS consistently outperforms all baselines across all training budgets. Notably, with very small datasets (around $\sim 10$ samples), the performance gap between Neural-HSS and the Green Learning model remains small. As the training size increases, this gap grows substantially in favor of Neural-HSS, highlighting its superior scalability in this setting.

**3D** The 3D setting is particularly challenging for neural PDE solvers, as generating large, high-fidelity 3D datasets is costly, not only in terms of simulation but also in data storage and model training. This makes it of paramount importance to have a model that can be trained with a very reduced number of samples when the underlying solution operator is structured.

As shown in the results in Figure 3, Neural-HSS consistently outperforms all baselines. With only 16 samples, Neural-HSS matches the performance of FNO trained on 64 samples and ResNet trained on 128 samples. At 32 samples, Neural-HSS achieves performance comparable to FNO trained on 256 samples. Training time is a crucial factor Figure 3. Neural-HSS trains significantly faster than both FNO and ResNet, completing training in about two and a half hours, compared to six hours for FNO and nearly one day and eighteen hours for ResNet. DeepONet trains even faster, requiring only about one hour, but it needs a much larger training set to achieve comparable performance. With a training size of only 256 samples, DeepONet cannot match Neural-HSS's accuracy. Since generating new samples is highly expensive, Neural-HSS is overall the most efficient choice, also in terms of time.

Together with the results in Table 2, this shows that the choice of higher-order HSS layer for $m-$dimensional problems proposed in Equation (1) is both simple and effective, even with very small outer rank values $r_{out}$, which for this experiment was set to two (see Appendix N).

### 3.2 ADDITIONAL EVALUATION OF PDE LEARNING PERFORMANCE

**1D PDEs.** To showcase the effectiveness of Neural-HSS on time-dependent problems, we train the models to predict the dynamics of the Heat and Burgers' equations. Specifically, the models learn the time-stepping operator $\mathcal{G} : u_t \mapsto u_{t+\delta t}$. At this timescale, prior work has shown that learning the residual yields better performance than direct prediction (Li et al., 2022). Following this strategy, our model $M$ is trained to predict $\delta u = u_{t+\delta t} - u_t$, so that, during inference, the state is updated as $u_{t+\delta t} = u_t + M(u_t)$. For training stability, we normalize the residuals by the maximum value in the

| Equation | Neural-HSS | | FNO | | ResNet | | DeepONet | | Green Learning | |
|---|---|---|---|---|---|---|---|---|---|---|
| | Params | Test Err | Params | Test Err | Params | Test Err | Params | Test Err | Params | Test Err |
| Heat Eq. | 45K | $3 \times 10^{-3}$ | 150K | $8 \times 10^{-3}$ | 165K | $1 \times 10^{-2}$ | 247K | $1 \times 10^{-2}$ | 83K | $1 \times 10^{-2}$ |
| Poisson Neumann BC | 37K | $2 \times 10^{-4}$ | 102K | $3 \times 10^{-3}$ | 165K | $4 \times 10^{-1}$ | 247K | $9 \times 10^{-3}$ | 83K | $3 \times 10^{-3}$ |
| Burgers' Eq. | 1.5M | 0.12 | 1.5M | 0.26 | 1.7M | 0.57 | 1.7M | 0.44 | - | - |

Table 1: Comparison between Neural-HSS and baseline models. We report the number of learnable parameters for each model and the test error Eq. (3).

training set, i.e., $\max \delta u$. We defer the model details with all the hyperparameters to Appendix N. We want to remark that we do not provide results for Green Learning on the Burgers equation, as the method requires the underlying PDE to be linear.

In both experiments, Neural-HSS outperforms the baselines while using fewer parameters, see Table 1. For the Heat equation, we observe trends consistent with the data efficiency experiments (Section 3.1). When trained on a larger dataset (approximately $10^4$ samples), Green Learning underperforms compared to both Neural-HSS and FNO. DeepONet is the most parameter-hungry model, we fix it at $247K$ parameters, since smaller variants consistently yielded weaker performance. We notice, moreover, that by the end of training, the parameter $\alpha$ of the LeakyReLU converged to 1, reflecting that the model has learned the underlying relation to be linear.

For the Burgers' equation, we find that employing a full-rank lifting and projection improves performance. This observation is aligned with findings in the literature on training models with low-rank parameter matrices, where the last and sometimes first layers are typically kept full-rank (Schotthöfer et al., 2022; Zangrando et al., 2024; Wang et al., 2021). Moreover, with respect to the heat equation, it is necessary to employ a higher rank in the intermediate layers to achieve sufficient expressivity. In this case, Neural-HSS consistently outperforms all other baselines while using fewer parameters and, similarly to the heat equation, the FNO emerges as the second-best performing.

We run another series of experiments on a modified version of the Burgers' equation, where, using a parameter $\beta$, we shift the equation towards the elliptic setting. While the other models seem to be unaffected by the change in the PDE setting, Neural-HSS shows a significant performance improvement as the equation approaches the elliptic setting, without any change to the hyperparameter setting. We defer to Appendix K the detailed results.

As shown in Figure 4, Neural-HSS's time for a single optimization step is comparable with other baseline models, while being significantly more effective as shown in Table 1. The efficiency is comparable to that of FNO, for which most of the computations are performed in Fourier space while truncating the modes, which significantly reduces cost. DeepONet is the fastest model; however, it also has the highest test error. Timings of only the inference phase are reported in Appendix L.

To further validate Neural-HSS, we tested all the methods on the Poisson equation with Neumann-type boundary conditions. Coherently with the theory, as it is known that the solution operator still has HSS structure even with these boundary conditions (Hackbusch, 2015), we observe that Neural-HSS achieves the lowest test error and parameter count. In this setting, ResNet underperforms with respect to Dirichlet boundary conditions, and the test error of all the other baselines are comparable.

**2D PDEs.** In Table 2 and Figure 5, we test model performance on 2D problems, when predicting the steady state of an equation or a specific time step. For the incompressible Navier-Stokes, we use a Z-score normalization as in the original papers (Yao et al., 2025; Huang et al., 2024); for the other equations, we use a max rescaling as for the 1D experiments. We defer the model details with all the hyperparameters to Appendix N. We do not provide results for Green Learning on the non-elliptic equation, as the method requires the underlying PDE to be linear. In addition to all the previous architectures, for the 2D case we can also compare with the FactFormer architecture proposed in (Li et al., 2023b).

For the Poisson equation, similar to the 1D case, smaller DeepONet architectures underperform. We clearly observe that the Neural-HSS significantly outperforms all baselines, with an even larger performance gap in the 2D experiments compared to the 1D case. As before, the FNO model ranks second, followed by the Green Learning model, consistent with the findings from the 1D experiments.

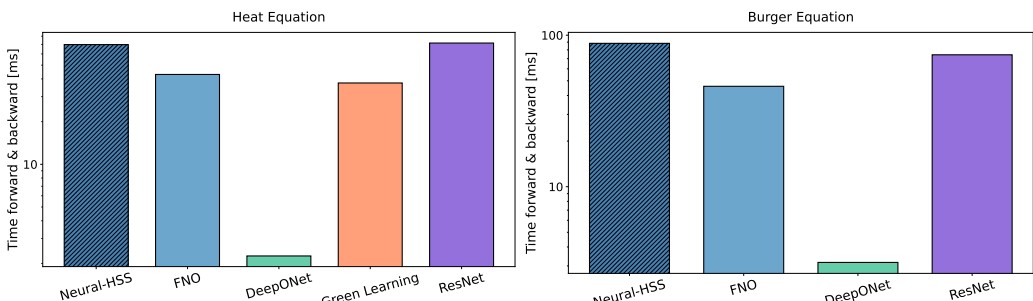

Figure 4: Timing of one forward+backward pass, calculated on two different datasets: **Left**: Heat equation. **Right**: Burgers' equation. We refer to Appendix L for inference only timings.

| | Neural-HSS | | FNO | | ResNet | | DeepONet | | Green Learning | | FactFormer | |
|---|---|---|---|---|---|---|---|---|---|---|---|---|
| Equation | Params | Test Err | Params | Test Err | Params | Test Err | Params | Test Err | Params | Test Err | Params | Test Err |
| Poisson Eq. | 37K | $7 \times 10^{-8}$ | 132K | $7 \times 10^{-6}$ | 165K | $3 \times 10^{-4}$ | 280K | $2 \times 10^{-2}$ | 83K | $5 \times 10^{-5}$ | 3.9M | $6 \times 10^{-3}$ |
| Poisson Eq. (Mixed BC) | 37K | $1 \times 10^{-4}$ | 132K | $6 \times 10^{-3}$ | 165K | $8 \times 10^{-2}$ | 280K | $4 \times 10^{-1}$ | 83K | $1 \times 10^{-2}$ | 3.9 M | $7 \times 10^{-2}$ |
| Poisson Eq. (L-shape domain) | 37K | $2 \times 10^{-2}$ | 132K | $2 \times 10^{-1}$ | 165K | $7 \times 10^{-1}$ | 280K | $8 \times 10^{-1}$ | 83K | $5 \times 10^{-1}$ | 3.9 M | $2 \times 10^{-1}$ |
| Helmholtz equation | 37K | $6 \times 10^{-3}$ | 132K | $5 \times 10^{-2}$ | 165K | $1 \times 10^{0}$ | 280K | $1 \times 10^{0}$ | 83K | $1 \times 10^{0}$ | 3.9 M | $7 \times 10^{-2}$ |
| Gray–Scott Eq. (Forward) | 329K | 0.294 | 2.1M | 0.331 | 1.99M | 0.273 | 2.3M | 0.315 | - | - | 3.9M | 0.291 |
| Gray–Scott Eq. (Inverse) | 329K | 0.203 | 2.1M | 0.208 | 1.99M | 0.193 | 2.3M | 0.276 | - | - | 3.9M | 0.192 |
| NS Eq. (Forward) | 329K | 0.123 | 2.1M | 0.483 | 1.99M | 0.481 | 2.3M | 0.409 | - | - | 3.9M | 0.14 |
| NS Eq. (Inverse) | 329K | 0.208 | 2.1M | 0.19 | 1.99M | 0.383 | 2.3M | 0.514 | - | - | 3.9M | 0.21 |

Table 2: Comparison between Neural-HSS and baseline models on 2D problems. We report the number of learnable parameters for each model and the test error Equation (2). With NS Eq., we refer to the incompressible Navier-Stokes equation.

In the non-elliptic setting, Neural-HSS demonstrates competitive performance compared to other models. This is particularly evident for the incompressible Navier–Stokes equation, where Neural-HSS outperforms all baselines in the forward problem. For the Gray–Scott forward model, Neural-HSS ranks as the second-best model, with ResNet achieving a slightly lower test error, but using significantly fewer parameters and fewer training steps. A very similar situation is observed in the Gray-Scott inverse problem, where FactFormer and ResNets' test errors are essentially equal and slightly lower than Neural-HSS. As for the 1D experiments, we use full-rank layers for the lift and the projection.

Also for these experiments, we report in Figure 5 the time required for a training step. In this setting, Neural-HSS remains highly competitive. In particular, for the Poisson equation, it is the second most computationally efficient model after DeepONets, but with a gap of six orders of magnitude in performance. For the Gray–Scott model and incompressible Navier–Stokes equation, training step timing is comparable to that of the FNO layer, which benefits from a low-cost forward pass due to mode truncation. The inverse Gray–Scott experiment highlights an interesting trade-off: although ResNet achieves the lowest test error, it requires more than one order of magnitude more in terms of time compared to Neural-HSS, which attains the second-lowest error. Given that the test-error gap between the two models is very small, Neural-HSS may be preferable in practice due to its substantially lower memory and computational cost.

Moreover, we remark here that the theory of HSS operators does not strictly depend on the domain being rectangular or on the specific Dirichlet boundary conditions. For non-rectangular domains, the Green's function remains smooth for well-separated points, so the same low-rank approximation property applies (Hackbusch, 2015). For this purpose, we experimentally demonstrate the performance of Neural-HSS on an L-shaped domain, for which we include the results in Table 2.
We also tested Neural-HSS on the 2D Poisson equation on the unit square domain deprived of a small disk of radius $R = 0.2$ in the center, with mixed boundary conditions. More precisely, on the external boundary of the square, we imposed periodic boundary conditions, while on the interior border of the disk, we imposed Neumann boundary conditions. While the exact structure of the solution operator in this more complicated geometry is not guaranteed theoretically to be well captured by the HSS structure, we show numerically that the inductive bias induced by Neural-HSS still works as a good approximation, as shown in the results in Table 2.

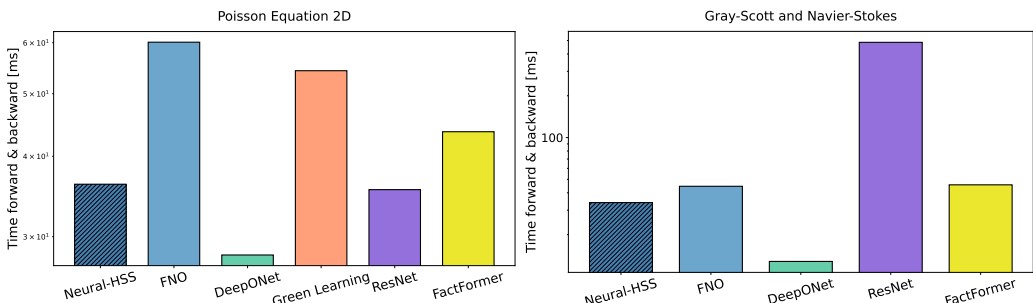

Figure 5: Timing of one forward+backward pass, calculated on two different datasets: **Left**: 2D Poisson Equation **Right**: Gray-Scott and Navier-Stokes. We refer to Appendix L for inference only timings.

Finally, compared to the one-dimensional experiments, we observe that the efficiency of Neural-HSS compared to the other baselines increases with the dimension as discussed in Section 2.1. We also refer to Section 3.1 for the three-dimensional case, in which this effect is even more evident.

## CONCLUSION

In this work, we present Neural-HSS, a novel hierarchical architecture inspired by the structure of the solution operator of Elliptic-type linear PDEs. Leveraging this very structured representation, we are able to produce lightweight neural PDE solvers with competitive performance with respect to state-of-the-art baselines and provable data-efficiency guarantees. The proposed architecture has an exact HSS structure for the one-dimensional case, while for higher-dimensional problems uses an approximate outer product expansion. Our model demonstrates superior scalability with respect to problem dimensionality compared to baseline approaches for data-efficient learning in the elliptic setting. Notably, the Green Learning model could not be trained in 3D at high resolution. Furthermore, in lower dimensions (1 and 2), our model also exhibits better scalability with respect to training set size, again outperforming all the baselines on linear and nonlinear PDEs.

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

## A   NOTATION AND USEFUL DEFINITION

**Definition A.1.** (Modal product)

Let $\mathcal{Z} \in \mathbb{R}^{d_1 \times \cdots \times d_m}, W \in \mathbb{R}^{D_k \times d_k}$. We denote with $(Z \bigtimes_k W) \in \mathbb{R}^{d_1 \times \ldots d_{k-1} \times D_K \times d_{k+1} \times \cdots \times d_m}$ the $k-$th modal product of $Z$ and $W$ as

$$(\mathcal{Z} \bigtimes_k W)_{i_1,\ldots,i_n} := \sum_{j_k=1}^{d_k} \mathcal{Z}_{i_1,\ldots,i_{k-1},j_k,i_{k+1},\ldots,i_m} W_{i_k,j_k}$$

**Definition A.2** (Cluster tree). Let $d \in \mathbb{N}$. A **cluster tree** is a perfect binary tree $\mathcal{T}$ that defines subsets of indices obtained by recursively subdividing $I = [1,\ldots,d]$.

The root $\gamma$ of the tree corresponds to the full index set $I$. Each non-leaf node $\tau$ is associated with a consecutive set of indices $I_\tau$, and its two children, $\alpha$ and $\beta$, correspond to two consecutive subsets $I_\alpha$ and $I_\beta$ such that $I_\tau = I_\alpha \cup I_\beta$ and $I_\alpha \cap I_\beta = \emptyset$.

If $\text{depth}(\mathcal{T}) = L$ and $d = 2^L k$ for $k \in \mathbb{N}$, we say that $\mathcal{T}$ is a **balanced cluster tree** if every non leaf node $\tau = [a,\ldots,b]$ is split exactly in the middle, i.e.

$$I_\alpha = [a, \tfrac{a+b}{2}], \qquad I_\beta = [\tfrac{a+b}{2}+1, b].$$

**Definition A.3.** ($\eta$- strong admissibility condition (Hackbusch, 2015, Definition 4.9))

Let $(\Omega, \|\cdot\|)$ be a normed space. We say that two subdomains $X, Y \subseteq \Omega$ satisfy the $\eta$-strong admissibility condition if

$$\max(\text{diam}(X), \text{diam}(Y)) \leq \eta \, \text{dist}(X, Y)$$

where

$$\text{diam}(X) := \sup_{x,x' \in X} \|x - x'\|, \quad \text{dist}(X,Y) := \max\{\sup_{x \in X} \inf_{y \in Y} \|x - y\|, \sup_{y \in Y} \inf_{x \in X} \|x - y\|\}$$

**Definition A.4.** (Asymptotic-smoothness (Hackbusch, 2015, Definition 4.14))

We say that $k : \Omega \to \mathbb{R}$ is asymptoticall smoth if there exist constants $C_0, C_1 > 0$ and $\mu \geq 0$ such that for all multi-indices $\alpha$

$$|\partial^\alpha k(z)| \leq C_0 \, C_1^{|\alpha|} |\alpha|! \, |z|^{-\mu-|\alpha|}, \quad z \neq 0,$$

.

## B   PROOF OF THEOREM 2.2

Thanks to (Hackbusch, 2015, Lemma 4.29), we have the bound

$$\inf_{B:\text{rank}(B)=r} \|A - B\|_F \leq \|k - k^{(r)}\|_{L^2(\Omega_\tau - \Omega_{\tau'})} \|R_\tau\|_2 \|R_{\tau'}\|_2 := C\|k - k^{(r)}\|_{L^2(\Omega_\tau - \Omega_{\tau'})} \leq$$

$$\leq C|\Omega_\tau - \Omega_{\tau'}| \|k - k^{(r)}\|_{L^\infty(\Omega_\tau - \Omega_{\tau'})}$$

In order for the upper bound to be controlled by $\varepsilon$, we need $\|k - k^{(r)}\| \leq \frac{\varepsilon}{C}$. Thanks to (Hackbusch, 2015, Theorem 4.22) we have that for $m = r^{1/D}$

$$\|k - k^{(r)}\|_{L^\infty(\Omega_\tau - \Omega_{\tau'})} \leq c_1 \left(\frac{c_2' \text{diam}_\infty(\Omega_\tau)}{\text{dist}(\Omega_\tau, \Omega_{\tau'})}\right)^m \leq c_1(c_2'\eta)^m$$

Therefore we finally have

$$\inf_{B:\text{rank}(B)=r} \|A - B\|_F \leq Cc_1(c_2'\eta)^{r^{1/D}} \leq \varepsilon$$

Therefore, for $r = O(\log(1/\varepsilon)^D)$ the inequality is satisfied.

## C  PROOF OF THEOREM 2.3

*Proof.* First, we note that if we assume, without loss of generality, that $\max_i r_i = r_1 \geq r$, and we choose $H_1 = \mathcal{G}^{-1}$, set $H_j$ equal to the identity matrix for all $j \geq 2$, and take $\alpha_i = 1$ for every $i$, then we obtain

$$\sup_{x \in \mathbb{R}^d} \|\mathcal{N}_\theta(x) - \mathcal{G}^{-1}x\|_2 = 0.$$

Conversely, in order to satisfy $\mathcal{L}_N(\theta) = 0$, it is necessary that $\alpha_i = 1$ for each $i$. Under this condition, $\mathcal{N}_\theta$ can be expressed as a product of HSS matrices associated with the same cluster tree $\mathcal{T}$, which implies that $\mathcal{N}_\theta$ is itself an HSS matrix associated with $\mathcal{T}$, with HSS rank $\sum_{i=1}^{\ell} r_i$. By the results in (Levitt & Martinsson, 2024), such an HSS matrix can be recovered uniquely with high probability from $\mathcal{O}\left(\sum_{i=1}^{\ell} r_i\right)$ observations. Therefore, all global minimizers must satisfy $W_\ell \ldots W_1 = \mathcal{G}^{-1}$, $\alpha_i = 1$, and therefore the claim.

$\square$

## D  FORWARD PASS

We assume, for simplicity, that the input and output vectors are of equal length.

---

**Algorithm 1:** Forward pass of the Neural-HSS linear layer

---

**Input:** $x \in \mathbb{R}^n$, tree $\mathcal{T}$ with depth $L$, rank $r$
**Output:** $y \in \mathbb{R}^n$
**Function** HSSForward($x$, $\mathcal{T}$, $r$):
  **if** $\text{depth}(\mathcal{T}) > 0$ **then**
    // Split input into leaf partitions
    Split $x = [x_1, \ldots, x_{2^L}]$, with $x_i \in \mathbb{R}^{|\tau_i|}$ for each $i$ ;   // $\tau_i$: leaf index set
    // Apply projection $V$ to each block
    **for** $i = 1, \ldots, 2^L$ **do**
      $z_i \leftarrow W_i^{(V)} x_i$ ;          // weights $W_i^{(V)} \in \mathbb{R}^{r \times |\tau_i|}$
    // Recurse on compressed representation
    $z \leftarrow [z_1, \ldots, z_{2^L}]$;
    $y \leftarrow$ HSSForward()$\left(z, \mathcal{T}_{2r}^{(L-1)}, r\right)$;  // Balanced cluster tree $\mathcal{T}_{2r}^{(L-1)}$
    // Split recursive output into blocks
    Split $y = [y_1, \ldots, y_{2^L}]$, with $y_i \in \mathbb{R}^r$ for each $i$;
    // Reconstruct using $U$ and diagonal $D$ blocks
    **for** $i = 1, \ldots, 2^L$ **do**
      $y_i \leftarrow W_i^{(U)} y_i$ ;          // weights $W_i^{(U)} \in \mathbb{R}^{|\tau_i| \times r}$
      $y_i \leftarrow y_i + W_i^{(D)} x_i$ ;     // weights $W_i^{(D)} \in \mathbb{R}^{|\tau_i| \times |\tau_i|}$
    **return** $[y_1, \ldots, y_{2^L}]$;
  **else**
    // Base case: leaf transformation
    **return** $Wx$ ;             // weight $W \in \mathbb{R}^{n \times n}$

---

**Complexity analysis**  A key advantage of Neural-HSS lies in its efficient memory footprint and inference cost. Under the simplifying assumption that the number of leaf indices equals $2r$, the storage requirement of an HSSForward() operator scales as $O(nr)$, and the cost of a matrix–vector multiplication is likewise $O(nr)$, as shown in (Chandrasekaran et al., 2006).

## E  LIFTING FOR SYSTEM OF PDES

Several PDEs have input and output multiple channels: to adapt our architecture for these kinds of problems, one needs to specify a lifting and projection architectures as depicted in Figure 1. While

multiple choices are possible, we employed the simplest possible choice as a backbone and, as a final layer, a (low-rank) linear tensor map

$$\phi_{\mathcal{W}} : \mathbb{R}^{D_1 \times \cdots \times D_M} \to \mathbb{R}^{d_1 \times \cdots \times d_m}, \quad \phi_{\mathcal{W}}(\mathcal{Z})_{\underline{\alpha}} = \sum_{\underline{\beta}} \mathcal{W}_{\underline{\alpha}, \underline{\beta}} \mathcal{Z}_{\underline{\beta}},$$

$$\mathcal{W} = \sum_{i=1}^r c_i u_i^{(1)} \otimes \cdots \otimes u_i^{(m)} \otimes v_i^{(1)} \otimes \cdots \otimes v_i^{(M)}, \quad c_i \in \mathbb{R}, u_i^{(j)} \in \mathbb{R}^{d_j}, v_i^{(j)} \in \mathbb{R}^{D_j},$$

where $c_i, u_i^{(j)}, v_i^{(j)}$ are learnable parameters. This allows us to easily extend Neural-HSS for a system of vector PDEs.

The overall model architecture, including a potential lifting layer for a system of PDEs, is depicted in Figure 1.

## F    DATA SPECIFICATION

For complete transparency and full reproducibility of our results in this section, we provide all the details about the data generation. If we did not generate them, we also include the source and the procedure for their generation.

### F.1    DATASETS

**Poisson equation.**    The $n$-dimensional Poisson equation

$$-\nabla \cdot \big(a(x)\nabla u\big) = f,$$

models diffusion or conduction in heterogeneous media, where $a(x) > 0$ is a spatially varying diffusivity and $f$ an external source. Although linear, the discretized problem is usually ill-conditioned, making it difficult to solve numerically. The Poisson equation is used to model different physical phenomena such as heat conduction, porous flow, and electrostatics in nonhomogeneous materials. During our experiments, we kept $a = 1$ and we trained the models to represent the mapping $\mathcal{G} \colon f \mapsto u$. To assess the quality of the model, we used the relative $L^2$ error on the test set.

**Heat equation.**    The heat equation is a fundamental linear partial differential equation that describes the diffusion of heat over time. In one spatial dimension, it is written as

$$u_t = \kappa u_{xx},$$

where $\kappa$ denotes the thermal diffusivity. The equation smooths out spatial inhomogeneities by dissipating gradients, leading to a monotone decay of energy in the system. In our experiments, we fixed the diffusion coefficient at $\kappa = 0.0002$. The objective was to learn the temporal dynamics of the system by approximating the mapping from the current state to its future evolution, i.e., $\mathcal{G} \colon u_t \mapsto u_{t+\delta t}$. In our experiments, we fixed $\delta t = 0.8$. Model performance is evaluated by rolling it through time and calculating the $L^2$ trajectory error on the test set.

**Helmholtz equation**    Efficient simulation of acoustic, electromagnetic, and elastic wave phenomena is essential in many engineering applications, including ultrasound tomography, wireless communication, and geophysical seismic imaging. When the problem is linear, the wave field $u$ typically satisfies the Helmholtz equation

$$-\Delta u - \kappa^2 u = f,$$

where the wavenumber $\kappa = \frac{\omega}{c}$ denotes the ratio between the (constant) angular frequency $\omega$ and the propagation speed $c$. Numerical solution of the Helmholtz equation is particularly challenging due to its non-symmetric, indefinite operator and the highly oscillatory nature of its solutions.

**Burgers equation.**    The Burgers equation is a fundamental nonlinear partial differential equation that captures the competing effects of nonlinear convection and viscous diffusion. In one spatial dimension, it takes the form

$$u_t + u u_x = \nu u_{xx},$$

where $\nu$ denotes the kinematic viscosity. The viscous term $\nu u_{xx}$ regularises these shocks, but introduces thin internal layers that are numerically stiff. In our experiments, we fixed the viscosity coefficient at $\nu = 0.001$, and the objective was to learn the temporal dynamics of the system, that is, to approximate the mapping from the current state to its future evolution, more formally learn the mapping $\mathcal{G} \colon u_t \mapsto u_{t+\delta t}$, to evaluate the models we use the $L^2$ trajectory error on the test set.

**Incompressible Navier-Stokes equation.** We consider the incompressible Navier–Stokes equations expressed in terms of the vorticity $w = \nabla \times v$:

$$\partial_\tau w(c, \tau) + v(c, \tau) \cdot \nabla w(c, \tau) = \nu \Delta w(c, \tau) + q(c),$$
$$\nabla \cdot v(c, \tau) = 0.$$

Here $v(c, \tau)$ is the velocity field, $q(c)$ a forcing term, and $\nu$ the viscosity; in the dataset, the viscosity is fixed $\nu = 10^{-3}$. These equations describe nonlinear, incompressible flow and are widely used to study vorticity dynamics, turbulence, and coherent structures. We use the data from (Yao et al., 2025; Huang et al., 2024) and we conduct a similar series of experiments, we learned the mapping $\mathcal{G} \colon w_0 \mapsto w_{10}$ and the related inverse problem $\mathcal{G} \colon w_{10} \mapsto w_0$. The model performances are evaluated using the relative $L^2$ error on the test set.

**Gray–Scott model** The Gray–Scott equations describe a prototypical reaction–diffusion system involving two interacting chemical species, $A$ and $B$, whose concentrations vary in space and time. They are given by

$$\begin{cases} \dfrac{\partial A}{\partial t} = \delta_A \Delta A - AB^2 + f(1 - A), \\[2mm] \dfrac{\partial B}{\partial t} = \delta_B \Delta B + AB^2 - (f + k)B. \end{cases}$$

Here, the parameters $f$ and $k$ regulate the *feed* and *kill* rates, respectively: $f$ controls the rate at which $A$ is supplied to the system, while $k$ controls the rate at which $B$ is removed. The diffusion constants $\delta_A$ and $\delta_B$ determine the spread of the two species in space. We used the dataset provided by (Ohana et al., 2024), where each trajectory consists of 1000 steps. Rolling out a machine learning model over so many steps becomes impractical due to the accumulation of errors. Therefore, as suggested in (Ohana et al., 2024), a relevant task is to predict the final state in order to understand the long-term behavior of the two chemical species. To this end, we model the mapping $\mathcal{G} \colon (A_0, B_0) \mapsto (A_{1000}, B_{1000})$, and we also pass the model the input parameters $f$ and $k$. Similarly to the incompressible Navier-Stokes, we also made an experiment for the inverse problem. The model performance is evaluated using the relative $L^2$ error on the test set.

### F.2 Poisson Equation (1D, 2D & 3D)

#### F.2.1 1D (Data Efficiency)

We generate datasets of solutions to the one-dimensional Poisson equation with homogeneous Dirichlet boundary conditions. The spatial domain is discretized uniformly with 1024 grid points over $x \in [0, 1]$, with grid spacing $h = 1/1024$. For each sample, the right-hand side $f(x)$ is drawn from a truncated Fourier sine series with 20 random modes:

$$f(x) = \sum_{k=1}^{10} c_k \sin(2k\pi x), \qquad c_k \sim \mathcal{U}(0, 1),$$

with $f(x)$ set to zero at the first two and last two grid points to enforce the boundary conditions. The Poisson problem is discretized using a fourth-order central finite difference scheme for the second derivative. The resulting banded linear system (with five diagonals) is solved efficiently using a direct banded solver. We generate 1,000 samples for training and 1,000 for testing. For training and testing our model, we downsample to 256.

#### F.2.2 2D

We generate datasets of solutions to the two-dimensional Poisson equation with homogeneous Dirichlet boundary conditions. The spatial domain is discretized uniformly with $64 \times 64$ grid points

over $(x, y) \in [0, 1]^2$, with grid spacing $h = 1/64$. For each sample, the right-hand side $f(x, y)$ is drawn from a truncated Fourier sine series with up to 10 random modes in each direction:

$$f(x, y) = \sum_{k_x=1}^{10} \sum_{k_y=1}^{10} c_{k_x,k_y} \sin(2\pi k_x x) \sin(2\pi k_y y) \qquad c_{k_x,k_y} \sim \mathcal{U}(0, 1),$$

where $k_x, k_y \in \{1, \ldots, 10\}$ are drawn uniformly at random for each sample. The forcing term $f(x, y)$ is set to zero along the boundary to enforce Dirichlet conditions.

The Poisson problem

$$-\Delta u(x, y) = f(x, y), \qquad u|_{\partial\Omega} = 0,$$

is discretized using a fourth-order finite difference scheme (nine-point Laplacian stencil), with a spatial resolution of $128 \times 128$. This resulting linear system is solved using a direct solver. We generate $4,200$ samples for training and $800$ for testing, and we downsample the spatial resolution to $64 \times 64$.

We further generate 2500 samples (2250 for training and 250 for testing) of the two-dimensional Poisson equation with homogeneous Dirichlet boundary conditions on a L-shape domain. We use the same strategy as the two-dimensional Poisson equation on the box to generate the data. We only change the domain shape, removing from the top right corner of the box a square with side $1/2$.

### F.2.3  3D (DATA EFFICIENCY)

We generate datasets of solutions to the three-dimensional Poisson equation with homogeneous Dirichlet boundary conditions. The spatial domain is discretized uniformly with $128^3$ grid points over $(x, y, z) \in [0, 1]^3$, with grid spacing $h = 1/n$ where $n \in \{128\}$. For each sample, the right-hand side $f(x, y, z)$ is constructed as a truncated Fourier sine series with randomly selected modes:

$$f(x, y, z) = \sum_{k_x=1}^{20} \sum_{k_y=1}^{20} \sum_{k_z=1}^{20} c_{k_x,k_y,k_z} \sin(k_x \pi x) \sin(k_y \pi y) \sin(k_z \pi z), \qquad c_{k_x,k_y,k_z} \sim \mathcal{U}(0, 1),$$

where $k_x, k_y, k_z \in \{3, \ldots, 23\}$ are drawn uniformly at random for each sample. The forcing term $f(x, y, z)$ is set to zero on the boundary of the domain to impose Dirichlet conditions.

The Poisson problem

$$-\Delta u(x, y, z) = f(x, y, z), \qquad u|_{\partial\Omega} = 0,$$

is discretized using a higher-order finite difference scheme based on a 19-point stencil, yielding a sparse linear system of size $n^3 \times n^3$. The system is solved using a sparse direct solver. We generate one dataset of 256 samples for training and 200 for testing.

### F.3  HEAT EQUATION

For the one-dimensional Heat equation, we generate 2000 trajectories (1800 for training and 200 for testing) with a time horizon $T = 8$ and time step $\delta t = 0.2$. The spatial domain is $X = [0, 1]$ with spatial resolution $\Delta x = 1/1023$ (1024 grid points) and homogeneous Dirichlet boundary conditions $u(0, t) = u(1, t) = 0$. The initial conditions are sampled from a truncated Fourier sine series with 10 modes and random coefficients $c_k \sim \mathcal{U}(0, 1)$:

$$u_0(x) = \sum_{k=1}^{10} 2 \cdot c_k \cdot \sin(k\pi x)$$

Each initial condition is normalized to have a maximum absolute value of 1. We use a second-order finite difference scheme for spatial discretization with diffusion coefficient $D = 0.0002$, and solve the resulting system of ODEs using the BDF (Backward Differentiation Formula) method with relative tolerance $10^{-4}$ and absolute tolerance $10^{-6}$. For training and testing our model, we downsample to 256.

### F.4 HELMHOLTZ EQUATION

We further generate 2500 samples (2250 for training and 250 for testing) of the two-dimensional Helmholtz equation with periodic boundary conditions on a box domain. We sample the forcing term from the same distribution as the one used in the Poisson equation. Moreover, we use the same grid resolution. We fix the $\kappa = 70$.

### F.5 BURGERS EQUATION

For the one-dimensional Burgers equation, we generate 2000 trajectories (1800 for training and 200 for testing) with a time horizon $T = 15.0$ and output time step $\delta t = 0.2$. The spatial domain is $X = [0, 1)$ with spatial resolution $\Delta x = 1/1024$ (1024 grid points) and Dirichlet boundary conditions. The initial conditions are sampled using a random Fourier series with 10 modes:

$$u_0(x) = \sum_{k=1}^{10} c_k \sin(2\pi(k+1)x)$$

where the coefficients $c_k \sim \mathcal{U}(-1, 1)$ are drawn from a uniform distribution. Each initial condition is normalized to have a maximum absolute value of 1. We employ second-order finite difference schemes for spatial discretization and solve the resulting system using the BDF (Backward Differentiation Formula) method with relative tolerances of $10^{-4}$ and absolute tolerances of $10^{-6}$. For training and testing our model, we downsample to 256.

### F.6 GRAY–SCOTT MODEL

We use the dataset from (Ohana et al., 2024), the follwoing are the datails for generating the data. Many numerical methods exist to simulate reaction–diffusion equations. If low-order finite differences are used, real-time simulations can be carried out using GPUs, with modern browser-based implementations readily available (Munafo, 2013; Walker et al., 2023). They choose to simulate with a high-order spectral method for accuracy and stability purposes. Specifically, they simulate equations (15)–(16) in two dimensions on the doubly periodic domain $[-1, 1]^2$ using a Fourier spectral method implemented in the MATLAB package Chebfun (Driscoll et al., 2014). The implicit–explicit exponential time-differencing fourth-order Runge–Kutta method (Kassam & Trefethen, 2005) is used to integrate this stiff PDE in time. The Fourier spectral method is applied in space, with linear diffusion terms treated implicitly and nonlinear reaction terms treated explicitly and evaluated pseudospectrally.

Simulations are performed using a $128 \times 128$ bivariate Fourier series over a time interval of 10,000 seconds, with a simulation time step size of 1 second. Snapshots are recorded every 10 time steps. The simulation trajectories are seeded with 200 different initial conditions: 100 random Fourier series and 100 randomly placed Gaussians. In all simulations, they set $\delta_A = 2 \times 10^{-5}$ and $\delta_B = 1 \times 10^{-5}$.

Pattern formation is then controlled by the choice of the "feed" and "kill" parameters $f$ and $k$. They choose six different $(f, k)$ pairs which result in six qualitatively different patterns, summarized in the following table:

| Pattern | $f$ | $k$ |
|---------|-------|-------|
| Gliders | 0.014 | 0.054 |
| Bubbles | 0.098 | 0.057 |
| Maze | 0.029 | 0.057 |
| Worms | 0.058 | 0.065 |
| Spirals | 0.018 | 0.051 |
| Spots | 0.030 | 0.062 |

On 40 CPU cores, it takes 5.5 hours per set of parameters, for a total of 33 hours across all simulations.

### F.7 NAVIER–STOKES EQUATION

We use the data provided by (Huang et al., 2024) for the two-dimensional Navier–Stokes equations. They follow this procedure to generate the data: The initial condition $w_0$ is sampled from a Gaussian

random field

$$w_0 \sim \mathcal{N}\big(0,\ 7^{1.5}(-\Delta + 49I)^{-2.5}\big).$$

The external forcing term is defined as

$$q(x_1, x_2) = \tfrac{1}{10}\big(\sin(2\pi(x_1 + x_2)) + \cos(2\pi(x_1 + x_2))\big).$$

We solve the Navier–Stokes equations in the stream-function formulation using a pseudo-spectral method. Specifically, the equations are transformed into the spectral domain via Fourier transforms, the vorticity equation is advanced in time in spectral space, and inverse Fourier transforms are applied to compute nonlinear terms in the physical domain. The system is simulated for $1$ second with $10$ time steps, and the vorticity field $w_t$ is stored at a spatial resolution of $128 \times 128$. In the dataset $\nu = 10^{-3}$ therefore the Reynolds number corresponding to $\mathrm{Re} = 1000$.

## G  METRICS

**Training Loss.**   As a training objective, we use the Mean Squared Error (MSE), defined as

$$\mathcal{L}(pred, target) = \frac{1}{|\mathcal{B}|}\sum_{b \in \mathcal{B}} ||pred_b - target_b||_2^2$$

**Evaluation Metrics.**   We assess the performance of our model using two different metrics. For steady-state problems or predictions at a specific time step, we employ the relative $L^2$ error, formally defined as

$$\mathcal{L}(pred, target) = \frac{1}{|\mathcal{B}|}\sum_{b \in \mathcal{B}} \frac{||pred_b - target_b||_2}{||target_b||_2}. \tag{2}$$

For temporal rollouts, in which the model is evaluated over a sequence of time steps, we adopt the $L^2$ trajectory error, mathematically expressed as

$$\mathcal{L}(pred, target) = \frac{1}{|\mathcal{B}|}\sum_{b \in \mathcal{B}} ||pred_b - target_b||_2. \tag{3}$$

## H   COMPLETE PLOT DATA EFFICIENCY POISSON 3D

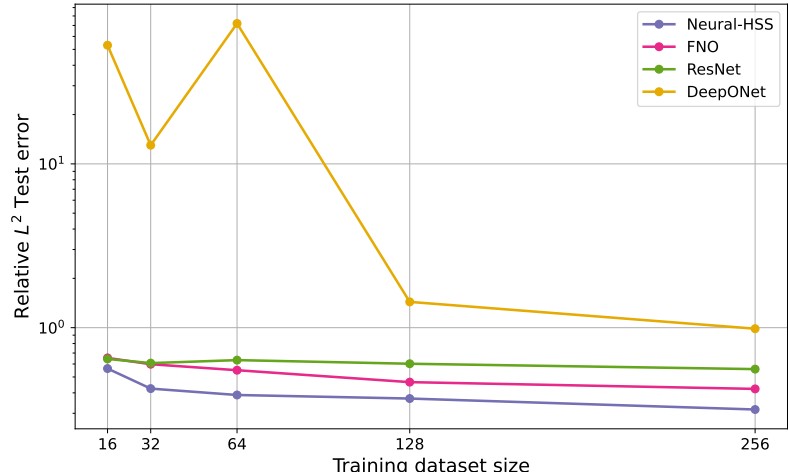

Figure 6: Train size vs relative test error for different models. The models are trained on a 3D Poisson equation

# I ABLATION ON HYPERPARAMETERS

In this section we include an ablation study on the three main hyperparameters of the Neural-HSS layer, namely rank (r), outer rank $r_{\text{out}}$ (for problems higher than $1D$), and levels $t$ of the hierarchical structure. We considered a grid of $r \in \{2, 4, 8, 12, 16\}$, $r_{\text{out}} \in \{2, 4, 8, 16, 24, 32\}$, $t \in \{1, 2, 3\}$, and for each one of the triplets $(r, r_{\text{out}}, t)$ of hyperparameters we trained a Neural-HSS model with those hyperparameters to predict the stationary state of the Gray-Scott problem. In Figure 7 we report the effect of the various couples of hyperparameters on the overall test performances. As we can observe in the plots, in the majority of the configurations, the model performs well, despite some small regions of high rank, outer rank, or levels in which the model started to overfit. Looking at the overall scale of the differences in performance, we can observe that the model is relatively stable with respect to all three hyperparameters, being the difference in terms of both test and train performance relatively small between the worst and best models observed.

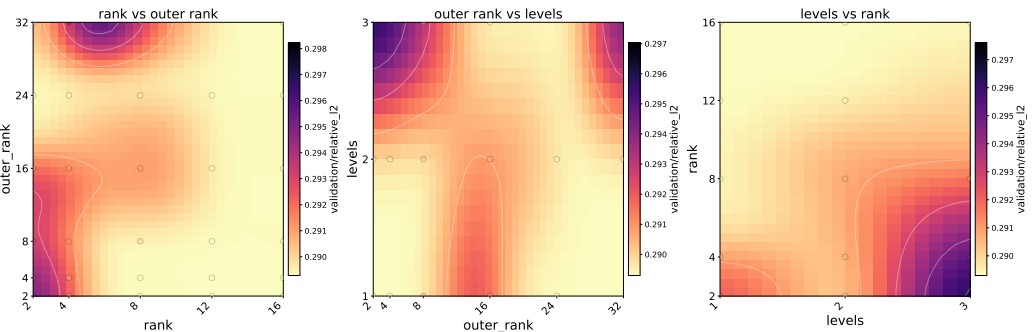

Figure 7: Ablation study on rank, outer rank, and levels for the Gray-Scott problem, where the color indicates the test error after training (rescaled from maximum to minimum observed, notice the scale in the colorbars).

# J TIME TO PERFORMANCE COMPARISON WITH NUMERICAL METHODS

In this section, we report a numerical experiment comparing the effectiveness of Operator learning techniques with respect to a standard numerical method to solve a one-dimensional Poisson equation with Neumann Boundary conditions and a two-dimensional Poisson equation with Dirichlet boundary conditions. Since the PDEs are linear, once one has the discretized Laplacian operator $L$ and the forcing term $f$, the problems can be simply solved using GMRES.

In Figure 8, we show a scatterplot of error against inference for different deep learning models and GMRES. As we can see in Figure 8, the computational time spent to approximate the solution is, at parity of error, more than one order of magnitude superior compared to the inference time of networks trained to fit the solution operator.

We further remark on another important case in which numerical methods cannot be used. In particular, when the differential operator $L$ is not known, but can be accessed just through the action of the inverse on a finite set of given forcing terms, which is the general setting of operator learning.

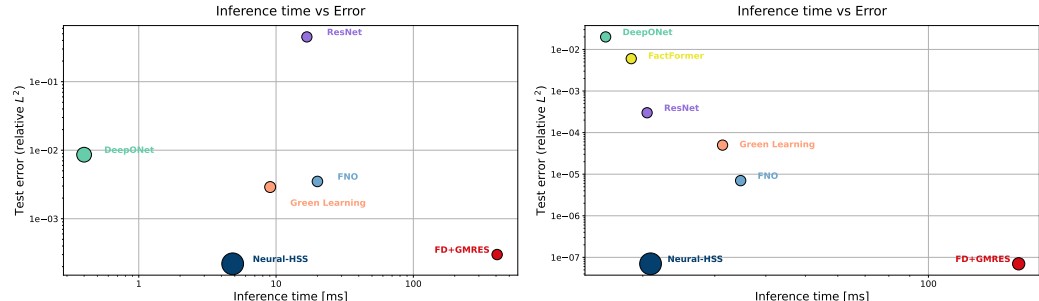

Figure 8: Scatter plot of forward time compared to relative $L^2$ test error $\rho$. The size of the points represents the ratio of performance and time. GMRES tolerance for termination was set to the one of the best performing model. **Left**: 1D Poisson with Neumann boundary conditions. **Right**: 2D Poisson equation with Dirichlet boundary conditions.

## K    ADDITIONAL EXPERIMENTS ON BURGERS'

We run the following series of experiments using $\beta \in \{1, 10^{-2}, 10^{-4}\}$ on the modified Burgers' equation

$$u_t + \beta u u_x = \nu u_{xx}.$$

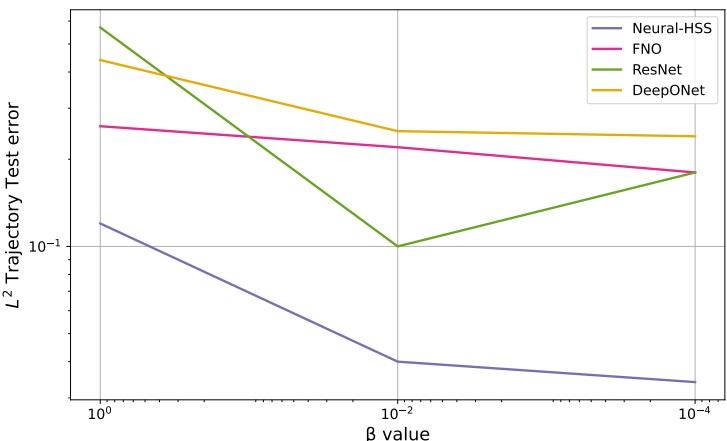

Figure 9: Trajectory test loss varying $\beta$

We adopt the same experimental setting used for the standard Burgers' equation, generating the data in the same manner and keeping the hyperparameters unchanged. When $\beta = 0$, the equation reduces to the heat equation, which is elliptic. The ResNet shows improved performance as $\beta$ decreases from 1 to $10^{-2}$, but its accuracy deteriorates at $\beta = 10^{-4}$. In contrast, the performance of FNO and DeepONet remains nearly constant across this range. The only model that benefits from the transition from the parabolic to the elliptic regime is Neural-HSS, highlighting its effectiveness in learning from data generated by elliptic PDEs.

## L  TIMING

In this section we will present further timing results. In particular, the setting is the same of Figures 4 and 5, for which we present the time of a training step for non-embedding layers with the same batch size used in the experiments (see Appendix N). In Figure 10, we present results in the same setting but just for the inference time.

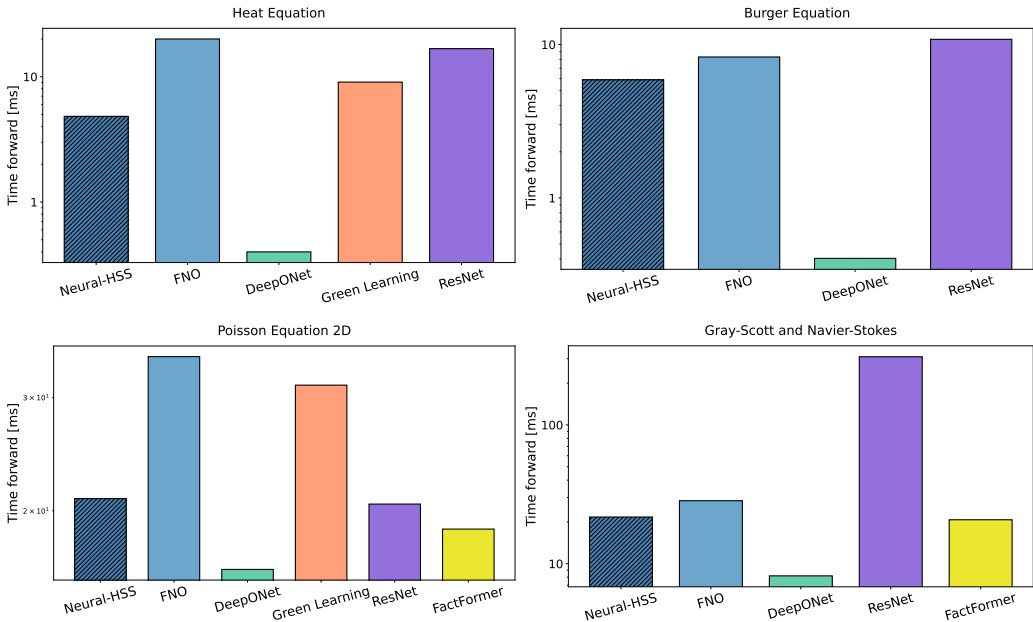

Figure 10: Timing of one forward pass, calculated on different datasets.

## M  DATA VISUALIZATION

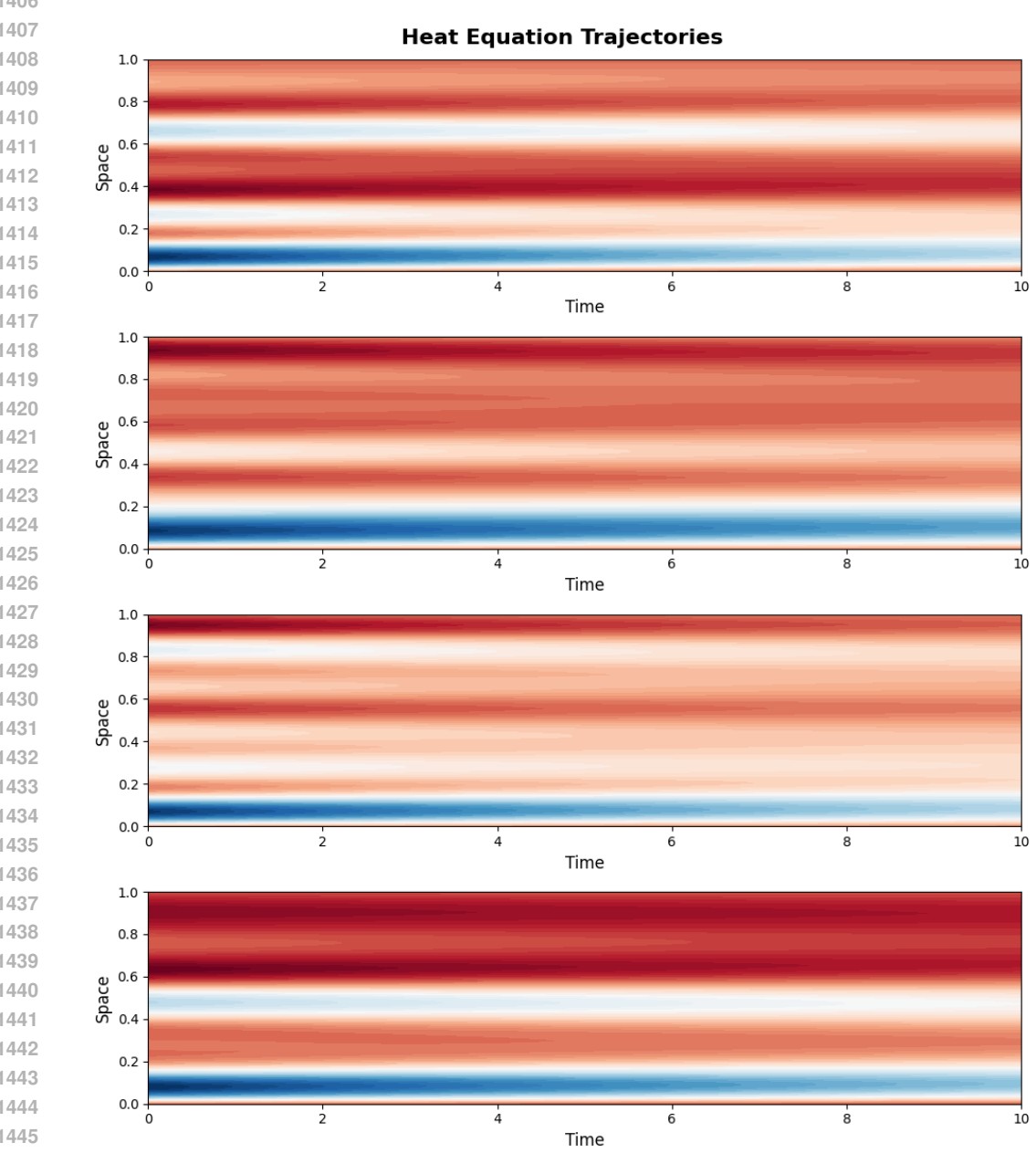

Figure 11: Example trajectories from the Heat equation dataset.

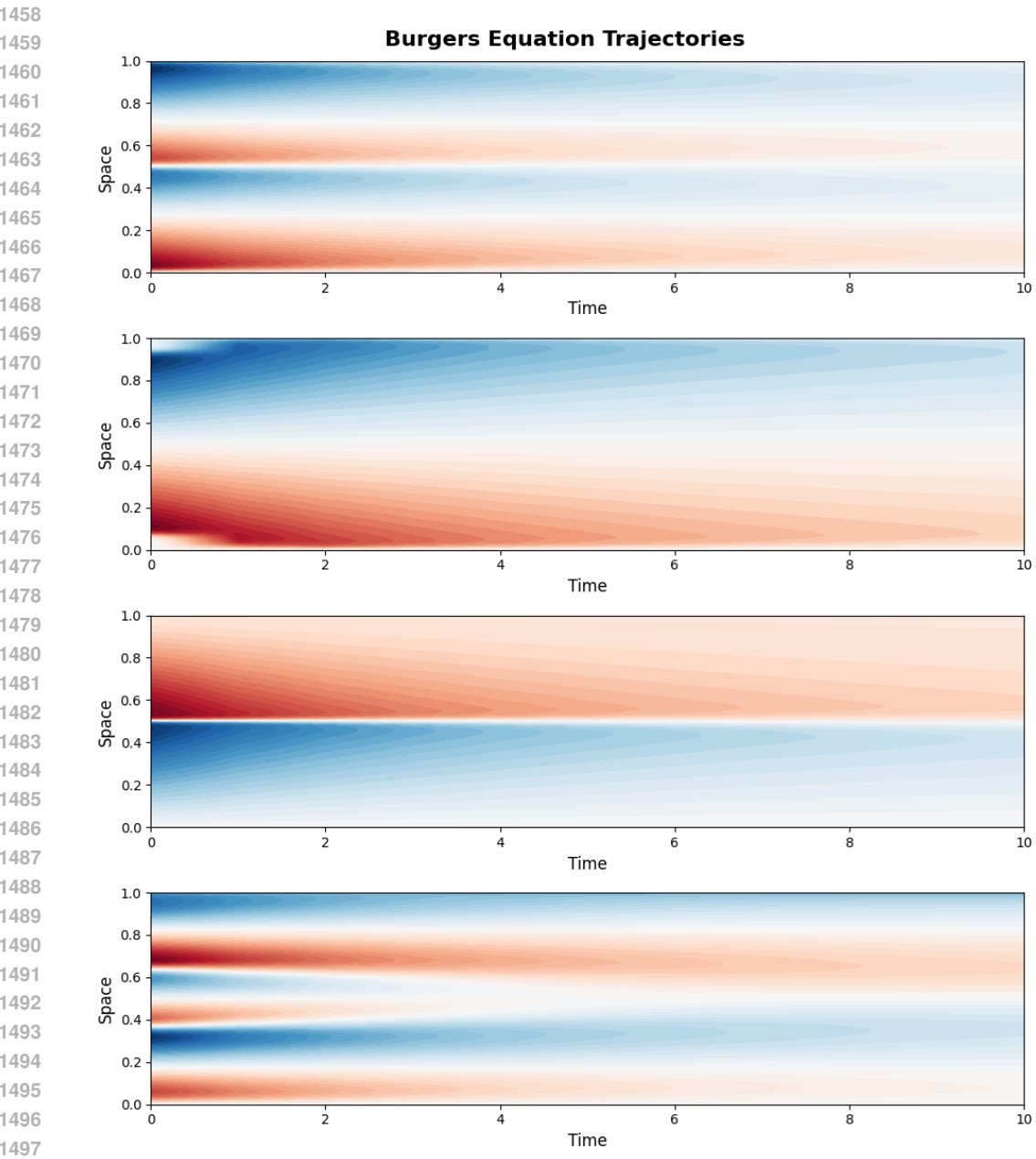

Figure 12: Example trajectories from the Burgers' equation dataset.

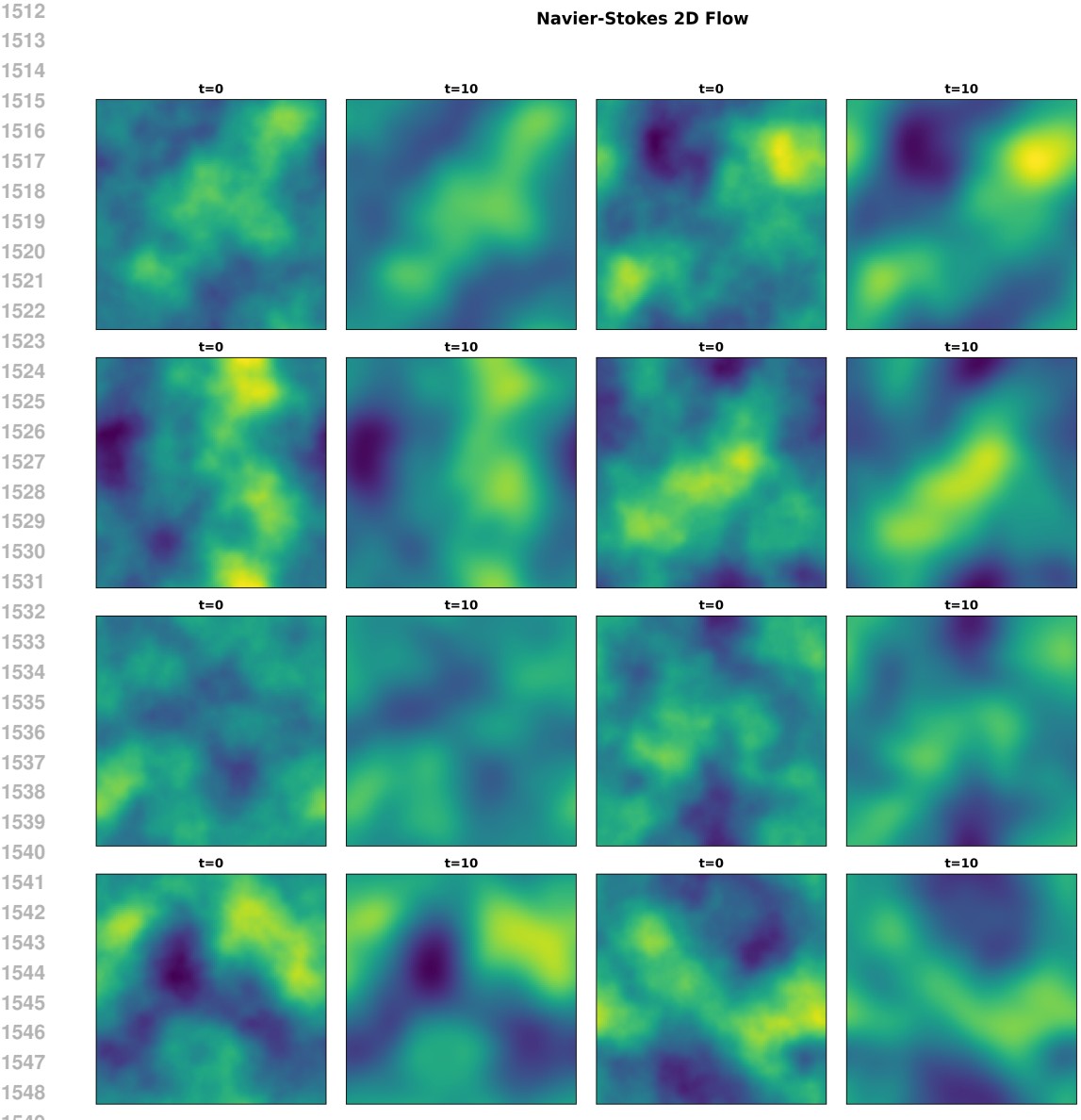

Figure 13: Example of snapshot from the Navier–Stokes equation dataset.

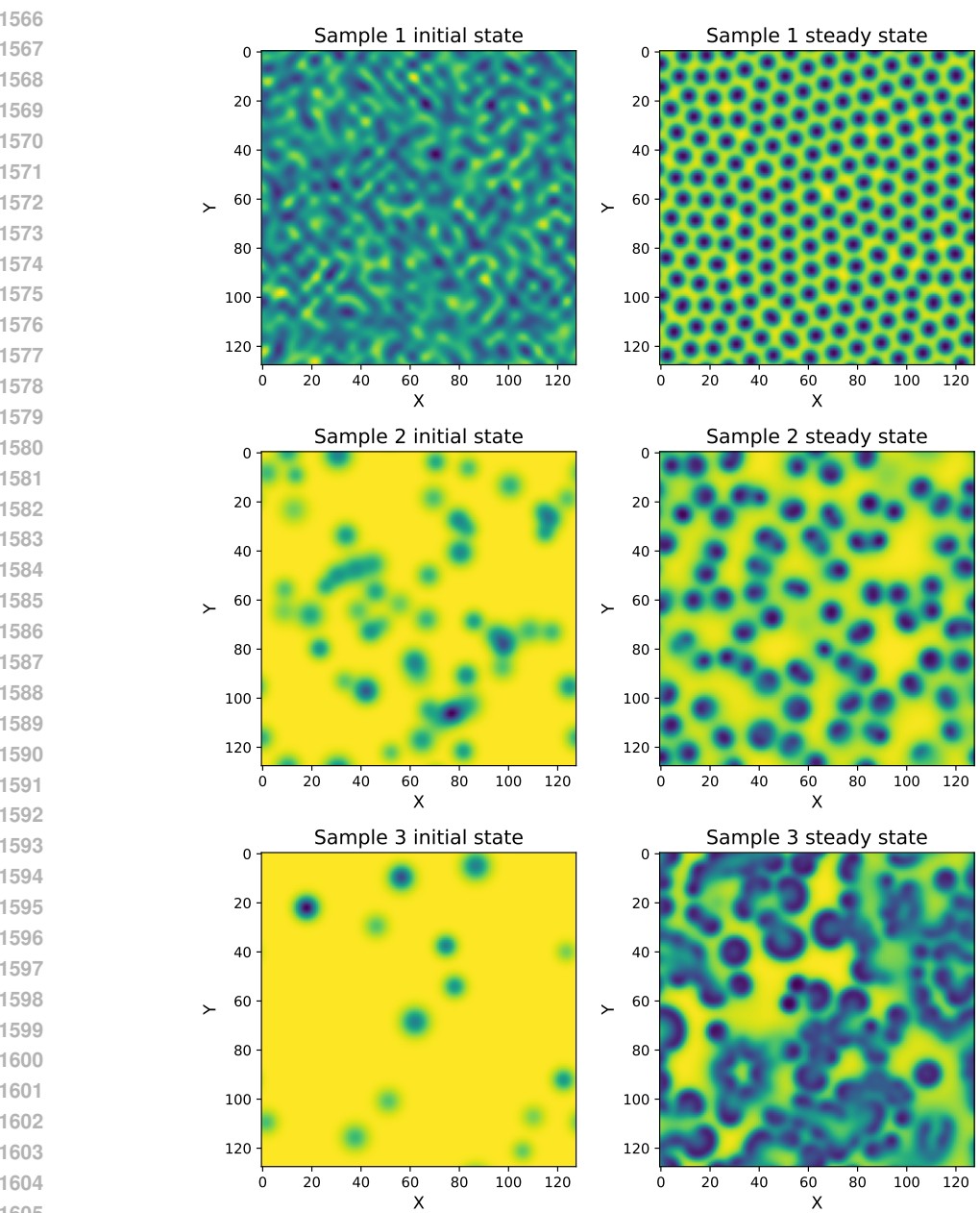

Figure 14: Gray-Scott data: input and steady state.

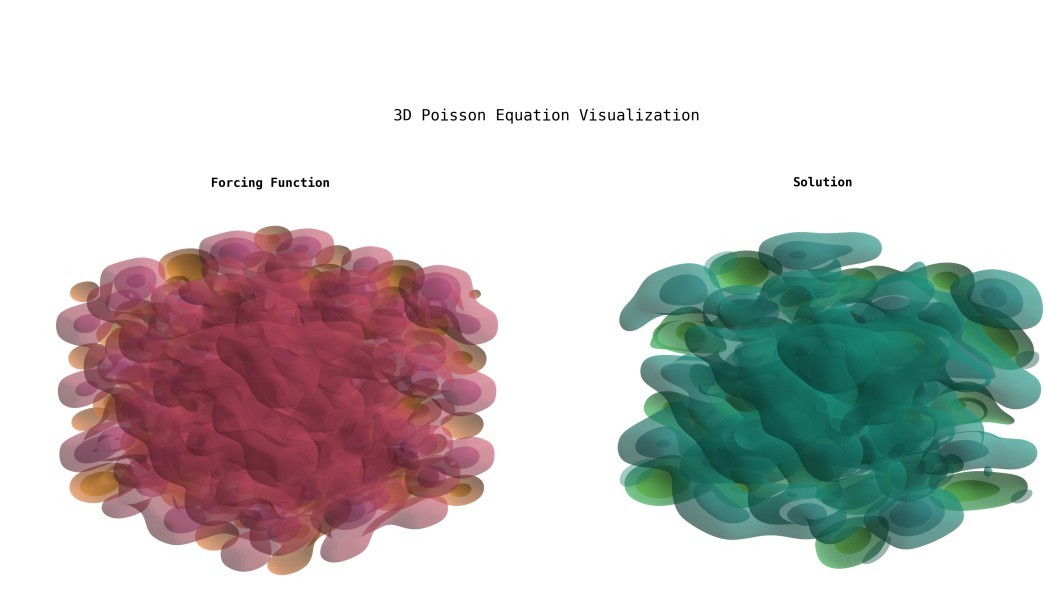

Figure 15: Example of a snapshot from the Poisson 3D equation dataset.

Figure 16: Example of a snapshot from the Poisson 3D equation dataset.

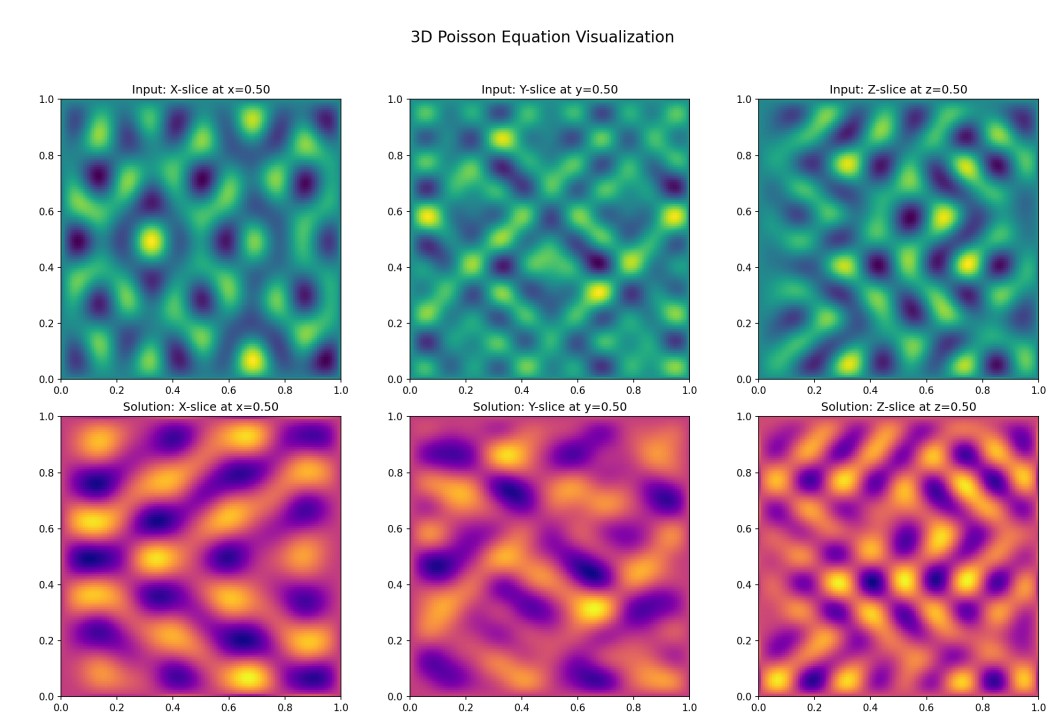

Figure 17: Example of 2 slice of a snapshot from the Poisson 3D equation dataset.

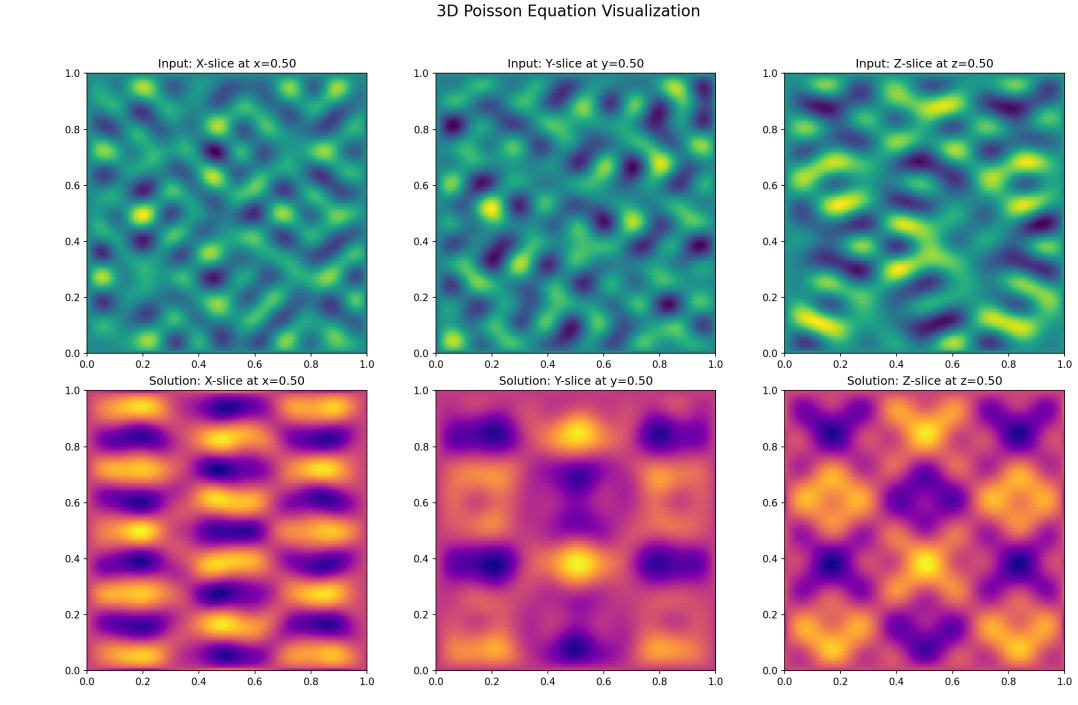

Figure 18: Example of 2 slice of a snapshot from the Poisson 3D equation dataset.

# N  MODEL & TRAINING DETAILS

| Configuration | Neural-HSS | FNO | ResNet | DeepONet | Green Learning |
|---|---|---|---|---|---|
| Depth | 3 | 6 | 12 | $4^{1}$ | 6 |
| Embedding Dimension | 256 | 32 | 48 | 192 | 128 |
| Activation | Trainable LeakyReLU | GeLU | GeLU | GeLU | Rational activation [2] |
| Levels | 3 | - | - | - | - |
| Rank | 2 | - | - | - | - |
| Modes | - | 12 | - | - | - |
| Peak learning rate | $1 \times 10^{-3}$ | $1 \times 10^{-3}$ | $1 \times 10^{-3}$ | $1 \times 10^{-3}$ | $1 \times 10^{-3}$ |
| Minimum learning rate | $1 \times 10^{-5}$ | $1 \times 10^{-5}$ | $1 \times 10^{-5}$ | $1 \times 10^{-5}$ | $1 \times 10^{-5}$ |
| Weight decay | $1 \times 10^{-3}$ | $1 \times 10^{-3}$ | $1 \times 10^{-3}$ | $1 \times 10^{-3}$ | $1 \times 10^{-3}$ |
| Learning rate schedule | Cosine | Cosine | Cosine | Cosine | Cosine |
| Optimizer | AdamW | AdamW | AdamW | AdamW | AdamW |
| $(\beta_1, \beta_2)$ | $(0.9, 0.99)$ | $(0.9, 0.99)$ | $(0.9, 0.99)$ | $(0.9, 0.99)$ | $(0.9, 0.99)$ |
| Gradient clip norm | 1 | 1 | 1 | 1 | 1 |
| Epochs | 1000 | 1000 | 1000 | 1000 | 1000 |
| Batch size | 256 | 256 | 256 | 256 | 256 |

Table 3: Models and training configuration for the **Heat Equation** and **Poisson Equation** with Neumann boundary conditions.

| Configuration | Neural-HSS | FNO | ResNet | DeepONet |
|---|---|---|---|---|
| Depth | 4 | 6 | 12 | 4 |
| Embedding Dimension | 1024 | 64 | 150 | 512 |
| Activation | Trainable LeakyReLU | GeLU | GeLU | GeLU |
| Levels | 3 | - | - | - |
| Rank | 32 | - | - | - |
| Modes | - | 32 | - | - |
| Peak learning rate | $5 \times 10^{-4}$ | $5 \times 10^{-4}$ | $5 \times 10^{-4}$ | $5 \times 10^{-4}$ |
| Minimum learning rate | $1 \times 10^{-6}$ | $1 \times 10^{-6}$ | $1 \times 10^{-6}$ | $1 \times 10^{-6}$ |
| Weight decay | $1 \times 10^{-3}$ | $1 \times 10^{-3}$ | $1 \times 10^{-3}$ | $1 \times 10^{-3}$ |
| Learning rate schedule | Cosine | Cosine | Cosine | Cosine |
| Optimizer | AdamW | AdamW | AdamW | AdamW |
| $(\beta_1, \beta_2)$ | $(0.9, 0.99)$ | $(0.9, 0.99)$ | $(0.9, 0.99)$ | $(0.9, 0.99)$ |
| Gradient clip norm | 1 | 1 | 1 | 1 |
| Epochs | 1500 | 1500 | 1500 | 1500 |
| Batch size | 256 | 256 | 256 | 256 |

Table 4: Models and training configuration for the **Burgers' Equation**.

---

[1] The depth is the same for the Trunk network and the Branch network
[2] (Boullé et al., 2020)

| Configuration | Neural-HSS | FNO | ResNet | DeepONet | Green Learning | FactFormer |
|---|---|---|---|---|---|---|
| Depth | 3 | 4 | 4 | $4^3$ | 6 | 1 |
| Embedding Dimension | 64 | 16 | 64 | 64 | 64 | 384 |
| Activation | Trainable LeakyReLU | GeLU | GeLU | GeLU | Rational activation[4] | GeLU |
| Levels | 2 | - | - | - | - | - |
| Rank | 2 | - | - | - | - | - |
| outer rank | 8 | - | - | - | - | - |
| Modes | - | 8 | - | - | - | - |
| Head dim | - | - | - | - | - | 8 |
| Head Num | - | - | - | - | - | 1 |
| Kern mult | - | - | - | - | - | 1 |
| Peak learning rate | $8 \times 10^{-4}$ | $8 \times 10^{-4}$ | $8 \times 10^{-4}$ | $8 \times 10^{-4}$ | $8 \times 10^{-4}$ | $8 \times 10^{-4}$ |
| Minimum learning rate | $1 \times 10^{-5}$ | $1 \times 10^{-5}$ | $1 \times 10^{-5}$ | $1 \times 10^{-5}$ | $1 \times 10^{-5}$ | $1 \times 10^{-5}$ |
| Weight decay | $1 \times 10^{-5}$ | $1 \times 10^{-5}$ | $1 \times 10^{-5}$ | $1 \times 10^{-5}$ | $1 \times 10^{-5}$ | $1 \times 10^{-5}$ |
| Learning rate schedule | Cosine | Cosine | Cosine | Cosine | Cosine | Cosine |
| Optimizer | AdamW | AdamW | AdamW | AdamW | AdamW | AdamW |
| $(\beta_1, \beta_2)$ | (0.9, 0.99) | (0.9, 0.99) | (0.9, 0.99) | (0.9, 0.99) | (0.9, 0.99) | (0.9, 0.99) |
| Gradient clip norm | 1 | 1 | 1 | 1 | 1 | 1 |
| Epochs | 500 | 500 | 500 | 500 | 500 | 500 |
| Batch size | 128 | 128 | 128 | 128 | 128 | 128 |

Table 5: Models and training configuration for the **2D Poisson Equation** and **2D Poisson Equation (L-shape domain)** .

| Configuration | Neural-HSS | FNO | ResNet | DeepONet | Green Learning | |
|---|---|---|---|---|---|---|
| Depth | 3 | 4 | 4 | $4^5$ | 6 | 1 |
| Embedding Dimension | 64 | 16 | 64 | 64 | 64 | 384 |
| Activation | Trainable LeakyReLU | GeLU | GeLU | GeLU | Rational activation[6] | GeLU |
| Levels | 2 | - | - | - | - | - |
| Rank | 2 | - | - | - | - | - |
| outer rank | 8 | - | - | - | - | - |
| Modes | - | 8 | - | - | - | - |
| Head dim | - | - | - | - | - | 8 |
| Head Num | - | - | - | - | - | 1 |
| Kern mult | - | - | - | - | - | 1 |
| Peak learning rate | $8 \times 10^{-4}$ | $8 \times 10^{-4}$ | $8 \times 10^{-4}$ | $8 \times 10^{-4}$ | $8 \times 10^{-4}$ | $8 \times 10^{-4}$ |
| Minimum learning rate | $1 \times 10^{-5}$ | $1 \times 10^{-5}$ | $1 \times 10^{-5}$ | $1 \times 10^{-5}$ | $1 \times 10^{-5}$ | $1 \times 10^{-5}$ |
| Weight decay | $1 \times 10^{-5}$ | $1 \times 10^{-5}$ | $1 \times 10^{-5}$ | $1 \times 10^{-5}$ | $1 \times 10^{-5}$ | $1 \times 10^{-5}$ |
| Learning rate schedule | Cosine | Cosine | Cosine | Cosine | Cosine | Cosine |
| Optimizer | AdamW | AdamW | AdamW | AdamW | AdamW | AdamW |
| $(\beta_1, \beta_2)$ | (0.9, 0.99) | (0.9, 0.99) | (0.9, 0.99) | (0.9, 0.99) | (0.9, 0.99) | (0.9, 0.99) |
| Gradient clip norm | 1 | 1 | 1 | 1 | 1 | 1 |
| Epochs | 1000 | 1000 | 1000 | 1000 | 1000 | 1000 |
| Batch size | 64 | 64 | 64 | 64 | 64 | 64 |

Table 6: Models and training configuration for the **Helmholtz equation** .

| Configuration | Neural-HSS | FNO | ResNet | DeepONet | FactFormer |
|---|---|---|---|---|---|
| Depth | 3 | 4 | 4 | 4 | 1 |
| Embedding Dimension | $128 \times 128$ | 32 | 64 | 128 | 384 |
| Activation | Trainable LeakyReLU | GeLU | GeLU | GeLU | GeLU |
| Levels | 2 | - | - | - | - |
| Rank | 8 | - | - | - | - |
| Outer Rank | 8 | - | - | - | - |
| Modes | - | 32 | - | - | - |
| Head dim | - | - | - | - | 8 |
| Head Num | - | - | - | - | 1 |
| Kern mult | - | - | - | - | 1 |
| Peak learning rate | $1 \times 10^{-3}$ | $1 \times 10^{-3}$ | $1 \times 10^{-3}$ | $1 \times 10^{-3}$ | $1 \times 10^{-3}$ |
| Minimum learning rate | $1 \times 10^{-5}$ | $1 \times 10^{-5}$ | $1 \times 10^{-5}$ | $1 \times 10^{-5}$ | $1 \times 10^{-5}$ |
| Weight decay | 0 | 0 | 0 | 0 | 0 |
| Learning rate schedule | Cosine | Cosine | Cosine | Cosine | Cosine |
| Optimizer | AdamW | AdamW | AdamW | AdamW | AdamW |
| $(\beta_1, \beta_2)$ | (0.9, 0.99) | (0.9, 0.99) | (0.9, 0.99) | (0.9, 0.99) | (0.9, 0.99) |
| Gradient clip norm | 1 | 1 | 1 | 1 | 1 |
| Epochs | 10 | 10 | 10 | 10 | 10 |
| Batch size | 64 | 64 | 64 | 64 | 64 |

Table 7: Models and training configuration for the **Incompressible Navier-Stokes and Gray-Scott**.

| Configuration | Neural-HSS | FNO | ResNet | DeepONet |
|---|---|---|---|---|
| Depth | 1 | 2 | 2 | 4 |
| Embedding Dimension | $128 \times 128 \times 128$ | 14 | 48 | 128 |
| Activation | Trainable LeakyReLU | GeLU | GeLU | GeLU |
| Levels | 2 | - | - | - |
| Rank | 4 | - | - | - |
| Outer Rank | 2 | - | - | - |
| Modes | - | 8 | - | - |
| Peak learning rate | $5 \times 10^{-3}$ | $5 \times 10^{-3}$ | $1 \times 10^{-4}$ | $1 \times 10^{-3}$ |
| Minimum learning rate | $1 \times 10^{-3}$ | $1 \times 10^{-3}$ | $1 \times 10^{-5}$ | $1 \times 10^{-5}$ |
| Weight decay | 0 | 0 | 0 | 0 |
| Learning rate schedule | Cosine | Cosine | Cosine | Cosine |
| Optimizer | AdamW | AdamW | AdamW | AdamW |
| $(\beta_1, \beta_2)$ | (0.9, 0.99) | (0.9, 0.99) | (0.9, 0.99) | (0.9, 0.99) |
| Gradient clip norm | 1 | 1 | 1 | 1 |
| Trainig steps | 16k | 16k | 16k | 16k |
| Batch size | 16 | 16 | 16 | 16 |

Table 8: Models and training configuration for the **data efficiency for 3D Poisson Equation**. We report the training steps since for each training size we match the training steps at 16k.

| Configuration | Neural-HSS | FNO | ResNet | DeepONet | Green Learning |
|---|---|---|---|---|---|
| Depth | 3 | 6 | 12 | $4^7$ | 6 |
| Embedding Dimension | 256 | 32 | 48 | 192 | 128 |
| Activation | Trainable LeakyReLU | GeLU | GeLU | GeLU | Rational activation [8] |
| Levels | 3 | - | - | - | - |
| Rank | 2 | - | - | - | - |
| Modes | - | 12 | - | - | - |
| Peak learning rate | $1 \times 10^{-3}$ | $1 \times 10^{-3}$ | $1 \times 10^{-3}$ | $1 \times 10^{-3}$ | $1 \times 10^{-3}$ |
| Minimum learning rate | $1 \times 10^{-5}$ | $1 \times 10^{-5}$ | $1 \times 10^{-5}$ | $1 \times 10^{-5}$ | $1 \times 10^{-5}$ |
| Weight decay | $1 \times 10^{-3}$ | $1 \times 10^{-3}$ | $1 \times 10^{-3}$ | $1 \times 10^{-3}$ | $1 \times 10^{-3}$ |
| Learning rate schedule | Cosine | Cosine | Cosine | Cosine | Cosine |
| Optimizer | AdamW | AdamW | AdamW | AdamW | AdamW |
| $(\beta_1, \beta_2)$ | (0.9, 0.99) | (0.9, 0.99) | (0.9, 0.99) | (0.9, 0.99) | (0.9, 0.99) |
| Gradient clip norm | 1 | 1 | 1 | 1 | 1 |
| Epochs | 500 | 500 | 500 | 500 | 500 |
| Batch size | 256 | 256 | 256 | 256 | 256 |

Table 9: Models and training configuration for the **data efficiency for 1D Poisson Equation**.

