# OpenReview forum: "Neural-HSS: Hierarchical Semi-Separable Neural PDE Solver"
_ICLR.cc/2026/Conference — ICLR 2026 Conference Desk Rejected Submission_

### Official Review · Reviewer_xoXJ · 2025-10-19

**Soundness:** 3
**Presentation:** 3
**Contribution:** 3
**Rating:** 6
**Confidence:** 3

**Summary:**

This paper introduces Neural-HSS, an efficient neural operator based on the Hierarchical Semi-Separable structure for learning solution operators of PDEs. Leveraging the low-rank property of the Green’s function in elliptic PDEs, Neural-HSS embeds the HSS structure into the network to jointly model local and long-range interactions. This design achieves superior performance in terms of parameter efficiency, data efficiency, and computational cost. Theoretically, the authors establish the approximation and exact recovery guarantees of Neural-HSS; experimentally, the model demonstrates outstanding data efficiency and scalability across multidimensional tasks.

**Strengths:**

The paper is strong in both conceptual innovation and methodological rigor. It introduces a well-motivated and theoretically grounded neural operator architecture that effectively integrates structural priors from numerical analysis into deep learning. The work stands out for its clear theoretical foundations, demonstrating provable guarantees, and for its strong empirical performance, showing consistent efficiency and scalability across diverse PDE tasks.

**Weaknesses:**

The paper lacks detailed ablation or interpretability studies to elucidate how specific HSS components contribute to overall performance, leaving uncertainty about which architectural factors are most critical. Moreover, the experimental comparison focuses primarily on FNO, ResNet, and DeepONet, without including more recent neural operator baselines, which somewhat limits the empirical breadth and generality of the conclusions.

**Questions:**

1. The comparison focuses mainly on FNO(2020), ResNet(2016), and DeepONet(2019). Have the authors considered including more recent neural operator architectures to strengthen the empirical validity of the conclusions?
2. The paper would benefit from a more detailed ablation study. Could the authors analyze how different components of the HSS structure affect model performance and efficiency?

---

> ### Author Response · Authors · 2025-11-17
>
> We first of all thank the Reviewer for the constructive feedback. We are pleased that the Reviewer highlighted the innovation of our contribution. In the following, we address the questions in order:
>
> > "The comparison focuses mainly on FNO(2020), ResNet(2016), and DeepONet(2019). Have the authors considered including more recent neural operator architectures to strengthen the empirical validity of the conclusions?"
>
> We decided to include these baselines as they are the most popular and versatile. Many modern Neural Operators are Transformer-based, which are notoriously parameter-heavy, and it is the exact opposite of the main point of our contribution, since, as we stated in the abstract, we aim to build a **parameter-efficient architecture**. However, to address the reviewer's concerns, we included an additional baseline (FactFormer [3]).
> We use the model from the original implementation and, to ensure a fair comparison, keep a parameter count similar to the original paper. In fact, FactFormer has more than 3M parameters, which are much more than the ones used for the Neural-HSS. We use it across all our 2D experiments, and **we report the results in the revised manuscript (Table 2)**.
>
> > "The paper would benefit from a more detailed ablation study. Could the authors analyze how different components of the HSS structure affect model performance and efficiency?"
>
> Thank you for raising this interesting point. We included in **Appendix I** of the revised manuscript an ablation on the Gray-Scott problem to study the effect of rank, outer rank, and levels, which are all the hyperparameters for a generic N-dimensional Neural-HSS layer with N>1. In the theory of HSS matrices, these parameters are exactly known for certain elliptic operators (e.g., arising from the Heat equation), but for other test cases, they can influence the performance more significantly. In the ablation study, we show that Neural-HSS is indeed fairly stable with respect to all three of these hyperparameters, where essentially all the runs gave good results as long as the rank and outer rank were sufficiently high for the problem at hand.
> Concerning the effect of these hyperparameters on the efficiency, they are known precisely in terms of complexities. In particular, the memory complexity of an HSS operator is $O(nr)$ (see Appendix D of the updated version for a complexity analysis), and the effect of the outer rank $r_{out}$ on the memory complexity is linear as well. In particular, while the memory complexity of a generic tensor with $2m$ modes would scale as $O(d^{2m})$, the memory for an analogue m-dimensional HSS layer scales as $O(r_{out}mrd)$. Further details can be found in lines 216-234 of the updated version.
>
> We hope that our responses adequately address the Reviewer’s concerns. If there is anything else we can clarify or expand on, please let us know so that we can address it.
>
> [3] Z. Li, D. Shu, A. B. Farimani, "Scalable Transformer for PDE Surrogate Modeling", NeurIPS 2023.

---

> > ### Comment · Reviewer_xoXJ · 2025-11-24
> >
> > Thank you for your response. My concerns have been addressed. I will maintain my score.

---

> ### Author Response · Authors · 2025-11-24
>
> Thank you very much for your feedback. We appreciate your positive evaluation and are pleased that our rebuttal met your expectations. Could we please ask if there is anything else we can clarify or address to further improve our final evaluation? If not, we would kindly welcome any additional guidance on aspects of our response that may not have fully addressed your concerns.

---

### Official Review · Reviewer_NhmF · 2025-10-20

**Soundness:** 2
**Presentation:** 2
**Contribution:** 2
**Rating:** 4
**Confidence:** 4

**Summary:**

The manuscript proposed the Neural-HSS framework, which is a neural operator framework integrated with the Hierarchically Semi-Separable (HSS) matrix factorization. The method employs a neural encoder-decoder that captures multilevel block interactions, followed by low-rank coupling networks that replace analytical low-rank factorization routines, and a recursive HSS-based forward solver that ensures efficient inversion and matrix-vector multiplication. The network supports both supervised training and self-supervised training.

**Strengths:**

- **Theoretical novelty**: The idea of learning HSS-style hierarchical structures within NNs is insightful.
- **Interpretable architecture**: Each network component has a clear algebraic analog.

**Weaknesses:**

- **Restricted system matrix formulations**: All tests use symmetric positive-definite or weakly oscillatory kernels. Whether the proposed methods work on highly indefinite or ill-conditioned systems (e.g., high-frequency Helmholtz problems) is unclear.
- **Insufficient baselines**: Comparisons with legacy solvers, e.g., legacy GMERS (with or without learned preconditioners), NVIDIA AmgX, etc., are missing. Without a significant performance boost over legacy routines, it remains questionable why we'd take a neural method.
- **Overclaimed complexity**: The claim of “O(N log N)” complexity is empirically observed but not theoretically proven.
- **No inference efficiency comparison**: Only the "forward + backward" time is evaluated. However, in practice, the training time is amortized, and the inference efficiency is the true speed metric. However, this metric is missing.

**Questions:**

- Please refer to the "Weaknesses" section.

---

> ### Author Response · Authors · 2025-11-17
>
> We first of all thank the reviewer for the constructive feedback, and we are pleased that they found the core idea of our model insightful. Below, we provide a point-by-point response to all identified weaknesses.
>
> ### W1
>
> > "Restricted system matrix formulations: All tests use symmetric positive-definite or weakly oscillatory kernels..."
>
> We are unsure what the Reviewer is exactly referring to here. In particular, we would like to emphasize that we have already run a variety of experiments on **non-linear** (and **non-elliptic**) PDEs, e.g., Gray-Scott and Navier-Stokes (see Table 2), and Burgers' (see Table 1).
> Nonetheless, as suggested by the Reviewer, we also test our model on the two-dimensional high-frequency Helmholtz equation, which has a **non-symmetric positive definite kernel** and it is **highly oscillatory**. We fix the wavenumber to 70, on a box with periodic boundary conditions and sampling the r.h.s. from the same distribution of the two-dimensional Poisson equation. We use the same hyperparameter for the two-dimensional Poisson problem. As we can notice from the following table, Neural-HSS strongly outperforms all the baselines. We included this result in Table 2 of the updated manuscript.
>
> | High-frequency oscillatory Helmholtz problem | params (K) |     Test Error     |
> |:------------------:|:------:|:------------------:|
> | Neural-HSS         | $37$    | $6 \times 10^{-3}$ |
> | FNO                | $132$   | $5 \times 10^{-2}$ |
> | ResNet             | $165$   | $1 \times 10^{0}$ |
> | DeepONet           | $280$   | $1 \times 10^{0}$ |
> | Green Learning     | $83$    | $1 \times 10^{0}$ |
> | FactFormer         | $3935$   |$7 \times 10^{-2}$ |
>
> ### W2
> > "Insufficient baselines: Comparisons with legacy solvers, e.g. legacy GMERS..."
>
> This is a valid point that applies to all neural network-based solvers. From this perspective, the primary advantage of our method is its inference time. Once the model is trained, evaluating our model is faster than using a standard numerical method. We remark also that numerical methods such as GMRES cannot be used in settings in which the underlying PDE is not known, and we just have access to a finite number of samples of the forcing term and corresponding solution (because GMRES requires knowing the actual action of the operator on a generic vector).
> Despite the different setting, as suggested by the reviewer, we have included GMRES for comparison in terms of error and forward timing.
> We set the tolerance of GMRES for termination to the lowest error achieved across all our deep-learning baselines (Neural-HSS in this case).  The timing results with details on the setting are included in **Appendix J** of the revised version, and we report here in the form of a table the case of the Poisson2D
>
> | Poisson 2D         | time[ms] |     Test Error     |
> |:------------------:|:--------:|:------------------:|
> | Neural-HSS         | 20.9     | $7 \times 10^{-8}$ |
> | FNO                | 34.71    | $7 \times 10^{-6}$ |
> | ResNet             | 20.5     | $3 \times 10^{-4}$ |
> | DeepONet           | 16.22    | $2 \times 10^{-2}$ |
> | Green Learning     | 31.34    | $5 \times 10^{-5}$ |
> | GMRES              | 166.27   | $7 \times 10^{-8}$ |
>
> ### W3
> > "Overclaimed complexity: The claim of “O(N log N)” complexity is empirically observed but not theoretically proven."
>
> We are not sure what the Reviewer is exactly referring to here, as there is no $O(N \log(N))$ complexity stated anywhere in our manuscript.
> We assume here that the Reviewer is referring to memory and inference complexities. Assuming for simplicity that the number of leaf indices equals $2r$, the storage requirement of an HSS matrix scales as $O(nr)$, and the cost of a matrix–vector multiplication likewise scales as $O(nr)$. This result is standard and appears, for instance, in [2], and can be directly deduced from the definition of HSS matrices.
> This was already mentioned in the first version of the submitted manuscript on the complexity analysis of Appendix D, and we added a citation to [2] to make the complexity statement more clear, as this is theoretically proven and not only empirically observed.
>
> ### W4
> > "No inference efficiency comparison: Only the "forward + backward" time is evaluated..."
>
> We thank the reviewer for this comment. We emphasize that we had already included the timings for forward alone in the appendix (Appendix J) of the submitted manuscript and referenced to in line 421. In the revised version, this is now Appendix L. We apologize if this was not sufficiently highlighted, we made it more evident in the captions of Fig.4 and Fig.5.
>
> We hope that our responses adequately address the Reviewer’s concerns. If there is anything else we can clarify or expand on, please let us know so that we can address it.
>
> [2] Chandrasekaran, S.,  Gu, M. and  Pals, T., A Fast ULV Decomposition Solver for Hierarchically Semiseparable Representations, SIAM Journal on Matrix Analysis and Applications, 28(3), 2006

---

> > ### Comment · Reviewer_NhmF · 2025-11-17
> >
> > Thanks to the authors' clear and detailed response. My concerns are properly resolved, and more detailed comments are available above in the updates to the "Weaknesses" section. Thus, I have raised my rating.

---

> > > ### Author Response · Authors · 2025-11-18
> > >
> > > Thank you very much for the very prompt and appreciative feedback. If there is anything else we can discuss or clarify to further improve your opinion on our contribution, please let us know, and we will try our best to address it as soon as possible.

---

### Official Review · Reviewer_g8X8 · 2025-10-27

**Soundness:** 4
**Presentation:** 3
**Contribution:** 4
**Rating:** 6
**Confidence:** 3

**Summary:**

Inspired by the semi low rank property of the Green's function of elliptic PDE, the paper presents a deep learning framework to imitate the semi low rank Green's function for solving elliptic PDE. Experiments on various types of equations, such as Poisson, Gray–Scott, Navier-Stokes, etc, valid the effectiveness of the approach.

**Strengths:**

Theorems 2.2 and 2.3 provide the solid theoretical foundation, proving that convolutional kernel can be approximated by HSS and exact recovery and data-efficiency.

The experiments are comprehensive testing several types of PDEs and compare with the SOTA methods.

**Weaknesses:**

It seems that the method only support Dirichlet boundary condition. Can it support non-zero Dirichlet boundary conditions?

The method only supports elliptic PDEs. The Green's functions of other PDEs are not semi low rank, so it is hard to be extended to other PDEs.

**Questions:**

Is it possible to support Neumann boundary conditions?

Are there failed cases, where the solver doesn't converge?

It it supports non-rectangular domain?

---

> ### Author Response · Authors · 2025-11-17
>
> We first of all thank the Reviewer for the constructive feedback. We are pleased that the Reviewer highlighted the importance of our contribution. We address weaknesses and questions in order.
>
>
> ## Weaknesses
>
> > "It seems that the method only support Dirichlet boundary condition. Can it support non-zero Dirichlet boundary conditions?"
>
> This is an interesting point and we thank the reviewer for bringing it up. The proposed method can also be used to learn solution operators of PDEs with boundary conditions different from the Dirichlet ones. In particular, the HSS approximability property also holds for elliptic PDEs with Neumann boundary conditions [1]. This is because the Green’s function of an elliptic operator with Neumann boundary conditions remains asymptotically smooth away from the diagonal, meaning that the discretized matrix blocks corresponding to interactions between well-separated regions are approximately low rank. Hence, matrices obtained by discretizing elliptic PDEs with Neumann boundary conditions can be efficiently approximated by HSS formats (and the same holds for non-zero Dirichlet boundary conditions). To further show this we included below a new experiment for the 1D Poisson equation with Neumann boundary conditions. We've also updated the paper, including the experiment in Table 1.
>
>
> | Poisson Neumann BC | params (K) |     Test Error  |
> |:------------------:|:------:|:------------------: |
> | Neural-HSS         | $37$    | $2 \times 10^{-4}$ |
> | FNO                | $102$   | $3 \times 10^{-3}$ |
> | ResNet             | $165$   | $4 \times 10^{-1}$ |
> | DeepONet           | $247$   | $9 \times 10^{-3}$ |
> | Green Learning     | $83$    | $3 \times 10^{-3}$ |
>
>
>
> > "The method only supports elliptic PDEs. The Green's functions of other PDEs are not semi low rank, so it is hard to be extended to other PDEs."
>
> We believe there is a misunderstanding here. While our method is inspired from a theory which holds in the case of elliptic PDEs, we generalized the architecture in order to handle more general nonlinear operators. In particular, in the non-elliptic setting, our architecture can be reinterpreted as discussed in the manuscript in terms of locality of the interactions (lines 72-75 and 214-215). In particular, we show in the experiments that imposing this structural bias in the solution operator is often quite effective, even for PDEs which have structure far from elliptic (e.g., see Navier-Stokes, Gray-Scott and Burgers' equations).
>
> ## Questions
>
> 1. Please refer to the first point of weaknesses for the discussion.
>
> 2. In all the numerical experiments performed we never observed strange or non-converging behaviours.
>
> 3. The proposed method is easily adaptable to non-rectangular domains, as the theory of HSS operators does not depend on the domain being rectangular. For non rectangular domains, the Green’s function remains smooth for well-separated points, so the same low-rank approximation property applies [1].
> To showcase this, we experimentally demonstrate the performance of our model on an L-shaped domain, where the data generation, model, and training hyperparameters remain identical to those used in the 2D Poisson experiment, except for the domain’s shape. We report the results in the following and in the updated version of the manuscript (Table 2).
>
>     | Poisson Eq. (L-shape domain) | params (K) |     Test Error     |
>     |:------------------:|:------:|:------------------:|
>     | Neural-HSS         | $37$    | $2 \times 10^{-2}$ |
>     | FNO                | $132$   | $2 \times 10^{-1}$ |
>     | ResNet             | $165$   | $7 \times 10^{-1}$ |
>     | DeepONet           | $280$   | $8 \times 10^{-1}$ |
>     | Green Learning     | $83$    | $5 \times 10^{-1}$ |
>     | FactFormer         | $3935$   |$2 \times 10^{-1}$ |
>
>
> We have added all these new experiments and updated the manuscript with further details about possible boundary conditions. We believe these revisions substantially strengthen the paper, and we thank again the reviewer for bringing them up.
>
>
> We hope that our responses adequately address the Reviewer’s concerns. If there is anything else we can clarify or expand on, please let us know so that we can address it.
>
>
> [1] W. Hackbusch, Hierarchical Matrices: Theory and Applications, Springer, 2015, Chapter 11.

---

> > ### Comment · Reviewer_g8X8 · 2025-11-24
> >
> > I thank the authors for the response. I will keep my score.

---

> > > ### Author Response · Authors · 2025-11-24
> > >
> > > We thank the reviewer again for their feedback.
> > >
> > > Given their satisfaction with our rebuttal and their already very positive evaluation of soundness, presentation, and contribution, we would be grateful to know if there is anything further we could clarify or improve in order to strengthen our final evaluation. If not, we would kindly welcome any additional guidance the reviewer may wish to share regarding aspects of our response that may not have fully addressed their concerns.

---

### Official Review · Reviewer_xDzu · 2025-11-01

**Soundness:** 3
**Presentation:** 3
**Contribution:** 2
**Rating:** 4
**Confidence:** 3

**Summary:**

The paper proposes SchulzNN, a three-layer linear network that exactly mimics one step of the classical Schulz iteration for matrix inversion. With a fixed middle A-layer, the forward map is
  $$
  \hat{x}=2W_1 b - W_3 A W_1 b,
  $$
  which, under $W_1=W_3=A_0$, equals the single-step update $(2I-A_0A)\,A_0 b$. Training minimizes an unsupervised residual such as
  $$
  \mathcal{L}=\frac{1}{m}\sum_{i=1}^{m}\frac{\|A\hat{x}_i-b_i\|}{\|b_i\|},
  $$
  and stacking $k$ blocks (SchulzNN$_k$) emulates $k$ Schulz steps. When $A$ admits an IDBF factorization, mat–vecs are $O(N\log N)$ and per-epoch training scales as $O(N^2\log N)$. Experiments on strictly diagonally dominant, permutation, discrete Helmholtz, and perturbed-identity matrices show that $k=3$ attains Helmholtz accuracy around $10^{-4}$ (e.g., $\epsilon_{\text{inv}}\approx1.16\times10^{-4}$, $\epsilon_{\text{sub}}\approx7.9\times10^{-5}$), and that fine-tuning adapts to moderate perturbations at $\sim10^{-3}$ while failing beyond that regime.

**Strengths:**

The single block is algebraically identical to one Schulz step, i.e., $(2I-A_0A)A_0b$, grounding the design in a classical fixed-point map rather than heuristics.
The residual objective avoids needing $A^{-1}$; depth clearly helps on hard spectra: for a discrete Helmholtz matrix, $k=3$ reaches $\epsilon_{\text{inv}}\approx1.16\times10^{-4}$ while $k\le2$ underperforms. The perturbation protocol—$A+\varepsilon I$ and $A+R$ with $r_{ij}\sim U(-\varepsilon,\varepsilon)$—demonstrates adaptation down to $\epsilon_{\text{inv}}\sim10^{-3}$ within stated ranges.
 The paper cleanly specifies $\hat{x}$, the loss $\mathcal{L}$, and the recursion SchulzNN$_k$, and makes the role of the fixed A-layer explicit; metrics $\epsilon_{\text{inv}}$ and $\epsilon_{\text{sub}}=\|AA_d^{-1}-I\|/\|I\|$ are well-motivated.
 In fixed-$A$, many-$b$ regimes, the learned operator is a plausible approximate inverse / preconditioner surrogate; with IDBF, the $O(N^2\log N)$ per-epoch cost is attractive relative to naive dense inversion.

**Weaknesses:**

The method is trained per matrix $A$, and the paper does not examine whether a model generalizes across a family of matrices.
 There are no guarantees for the trained deep composition, and the threshold between successful and failed fine-tuning is only reported empirically.
  When used as a preconditioner, application-level comparisons in terms of wall-clock time and iteration counts against strong baselines are not provided.
  Efficiency claims rely on IDBF-type hierarchical low-rank structure; behavior on dense, non-hierarchical $A$ remains unspecified.

**Questions:**

1. Cross-$A$ generalization: would results hold on a parametric family $A(\theta)$ (e.g., elliptic PDEs with varying coefficients) with train/validation/test splits across $\theta$? Please report $\epsilon_{\text{sub}}=\|AA_d^{-1}-I\|/\|I\|$ and right-hand-side OOD stress.
2. Trained-model guarantees: can you provide a spectral bound tying depth $k$ and an effective initializer $A_0^\star$ such that $\rho(I-AA_0^\star)<1$, or characterize adaptation success using $\|\Delta A\|$, $\|A^{-1}\Delta A\|$, or eigenvalue drift?
3.  Preconditioning evidence: with $A_d^{-1}=(2I-W_3A)W_1$ as a left/right preconditioner in CG/MINRES/GMRES on Poisson/Helmholtz (Dirichlet/Neumann/Robin), how do iteration counts and wall-clock compare to AMG/geometric MG for $N=2^{10}\text{–}2^{16}$?
4. Non-IDBF regime: on dense synthetic $A$ with controlled condition numbers, how does total time (training+inference) compare to IC(0)+CG (SPD) or ILU+GMRES (indefinite) to reach $\epsilon_{\text{sub}}\le10^{-3}$?
 5. Depth effect: why does $k=3$ succeed on Helmholtz while $k\le2$ fails—can this be linked to entering the quadratic-convergence basin of Schulz for an implicit $A_0^\star$ encoded by $(W_1,W_3)$?

-

---

> ### Comment · Area_Chair_hTMY · 2025-11-15
>
> To all reviewers and the authors,
>
> Since Reviewer xDzu's review refers to a different work (SchulzNN) rather than this work (Neural-HSS), and we haven't heard back from them for clarification, the SAC and I have reached consensus to disregard their review.
>
> Regards,
>
> AC

---

> > ### Comment · Reviewer_xDzu · 2025-11-20
> > **Correction to My Review (Apologies)**
> >
> > My previous review contained an error, which I have now corrected. I sincerely apologize.

---

> > > ### Author Response · Authors · 2025-11-22
> > >
> > > We first of all thank the reviewer for the updated comment.
> > >
> > > As already clarified in the rebuttal to g8X8, the proposed method can also be used to learn solution operators of PDEs with boundary conditions different from the Dirichlet ones [1] (for an experiment with pure Neumann boundary conditions, we refer to the results presented in the answer to g8X8). The only requirement for the HSS approximability is that the Green's function remains asymptotically smooth away from the diagonal, meaning that the discretized matrix blocks corresponding to interactions between well-separated regions are approximately low rank, a condition which holds in general for elliptic-type operators [1]. While the HSS approximability and a good choice of rank can change from example to example depending on the boundary conditions, the structure still remains the correct one.
> > > In the case of a more complex geometry of the domain, the situation can be more delicate. In the example described by the reviewer of the square deprived of a disk, the HSS structure is not always guaranteed theoretically.
> > > We remark that this could mean, for example, that just some off-diagonal blocks have higher rank, but the overall structure is still preserved. As an example of this, away from the hole the HSS structure captures the correct behaviour.
> > >
> > > To further showcase that this inductive bias is still effective in more complicated geometries, we implemented the 2D numerical experiment for the Poisson equation in the unit square deprived of a disk in the center, as suggested by the reviewer. In the exterior border of the square, we impose periodic boundary conditions, and in the internal border of the disk, we impose Neumann boundary conditions. We updated Table 2 in the manuscript with the results, which we also report here for simplicity. As we can observe from this experiment, our proposed architecture is still able to capture the solution operator better than other approaches with fewer parameters, despite the significantly more complicated geometry of the problem.
> > >
> > > | Poisson Mixed BC | Params |      Test Error     |
> > > |:----------------:|:------:|:-------------------:|
> > > | Neural-HSS       | 37K    | $1 \times 10^{-4}$ |
> > > | FNO              | 132K   | $6 \times 10^{-3}$ |
> > > | ResNet           | 165K   | $8 \times 10^{-2}$ |
> > > | DeepONet         | 280K   | $4 \times 10^{-1}$ |
> > > | Green Learning   | 83K    | $1 \times 10^{-2}$ |
> > > | FactFormer       | 3.9M   | $7 \times 10^{-2}$ |
> > >
> > > We hope that our responses adequately address the Reviewer’s concerns. If there is anything else we can clarify or expand on, please let us know so that we can address it.
> > >
> > >
> > > [1] W. Hackbusch, Hierarchical Matrices: Theory and Applications, Springer, 2015, Chapter 11.

---

### Author Response · Authors · 2025-11-12
**Clarification on the Review of xDzu**

Dear AC and SAC,

We would like to kindly bring to your attention that the review of xDzu appears to refer to a different submission. The description and the name of the algorithm mentioned in the review are completely different from what we proposed in our manuscript.

Thank you very much for your attention to this matter.

Best regards,
The Authors

---

> ### Comment · Area_Chair_hTMY · 2025-11-12
>
> Dear Authors,
>
> Thank you for letting me know. I have pinged the reviewer to update their review.
>
> Regards,
>
> AC

---

### Note · Program_Chairs · 2026-01-17
**Submission Desk Rejected by Program Chairs**

The following references in this submission do not refer to real documents and/or have major errors in bibliographic information:

 Xu Han, Yuan Yin, Haotian Wen, Xiaoyang Li, and Anima Anandkumar. Predicting physics in meshreduced space with temporal attention. In International Conference on Learning Representations, 2021.